# Histopathology of *Thecaphora frezzii* Colonization: A Detailed Analysis of Its Journey Through Peanut (*Arachis hypogaea* L.) Tissues

**DOI:** 10.3390/plants14071083

**Published:** 2025-04-01

**Authors:** María Florencia Romero, Sergio Sebastián Samoluk, José Guillermo Seijo, Ana María Gonzalez

**Affiliations:** 1Instituto de Botánica del Nordeste (UNNE-CONICET), Facultad de Ciencias Agrarias, Universidad Nacional del Nordeste, Corrientes 3400, Argentina; 2Instituto de Botánica del Nordeste (UNNE-CONICET), Facultad de Ciencias Exactas, Naturales y Agrimensura, Universidad Nacional del Nordeste, Corrientes 3400, Argentina; ssamoluk@exa.unne.edu.ar (S.S.S.); jgseijo@exa.unne.edu.ar (J.G.S.)

**Keywords:** biotrophic fungi, histopathology, host-pathogen interaction, peanut (*Arachis hypogaea*), peanut smut, teliosporogenesis, transpresorium, *Thecaphora frezzii*, vascular colonization

## Abstract

Over the past few decades, peanut smut, caused by *Thecaphora frezzii*, has evolved from an emerging disease to a major global threat to peanut production. However, critical knowledge gaps persist regarding the anatomical pathways and host responses involved in infection, colonization, and sporulation. This study examines the pathosystem and histopathology of the biotrophic phase of *T. frezzii* in the susceptible cv. Granoleico. Anatomical analyses were conducted using light microscopy, confocal laser scanning, and scanning electron microscopy. Our findings reveal that *T. frezzii* enters the host through the peg rather than the ovary tip, invading during the R2-subterranean phase. Fruit colonization occurs at the R3-stage when the mechanical layer between the mesocarp and endocarp has not yet formed. Hyphal entry into the seed takes place between the R3-medium and R3-late pod stages via the funiculus, leading to extensive seed coat colonization without penetrating the embryo. Once inside, hyperplasia and hypertrophy are triggered, coinciding with teliospore formation. Teliosporogenesis disrupts nutrient translocation, arresting embryo development. The hyphae colonize tissues intracellularly, utilizing living cells of the vascular bundles and following the peanut’s photoassimilate transport pathway. Investigating these structural responses in phenotypically contrasting peanut genotypes may provide key insights into the anatomical barriers and defense mechanisms that determine disease susceptibility and resistance, ultimately contributing to the development of resistant cultivars.

## 1. Introduction

The peanut (*Arachis hypogaea* L.) is a globally important crop, originally domesticated in South America [1,2,3,4,5], and is now grown in over 100 countries with an annual production exceeding 50 million tons in 2023. Due to its high protein (15–28%), oil (45–56%), and a variety of healthy micronutrients and bioactive compounds, peanuts provide the highest protein content among commonly consumed snack nuts and serve as a rich source of heart-healthy monounsaturated oil, contributing to their nutritional value and potential health benefits [6]. It is the third major oilseed in the world behind soybean (*Glycine max* (L.) Merr.) and cotton (*Gossypium hirsutum* L.) [7].

Since the early XXI century, peanut production has faced significant new challenges due to the emergence of a new devastating disease, peanut smut, caused by the fungus *Thecaphora frezzii* Carranza & Lindquist [8,9]. Symptoms of smut infection are not visually apparent on aboveground organs and become evident only after digging up and shelling peanut pods [10]. Pods of plants infected by *T. frezzii* may exhibit hypertrophy and abnormal shapes [10,11,12]; however, detailed histological studies describing the precise cells or tissues affected during these processes have not been conducted. Histopathological characterization has identified the peg as the primary entry point for hyphae and the vascular bundles as the conduit for invasion [8,11,12,13,14]. The most distinctive symptom is the production and accumulation of teliospores within the pods during later stages of infection, replacing the kernel tissue with a smutted mass [8,10,11,12] and significantly reducing peanut yield. To date, the specific kernel tissues or structures where teliosporogenesis initiates have not been described. 

*Thecaphora frezzii* is a soilborne fungus [8,9] originally described from the wild species *Arachis kuhlmanii* Krapov. & W.C. Gregory from Aquidauana, Mato Grosso, Brazil in 1962 [15], and more recently it was also identified in other wild species from Bolivia [16]. The disease did not appear in commercial peanut fields until 1995 when it was reported in Córdoba, Argentina [10]. Since the first detection, the disease has rapidly spread and it currently affects the entire peanut-producing region in Argentina, with pod incidence that can reach up to 70% in commercial fields [17]. The severity of this disease varies depending on environmental conditions and agricultural practices, and fields with the highest incidence (>50%) and severity show yield losses of up to 35% [9,18,19,20]. The mean incidence increased from 1.66% in 2015 to 11.47% in 2020 [17] and yield loss due to smut was estimated in 27,419 MT (US$14,151,800), representing 3.15% of total production [20].

The fungus has a monocyclic life cycle. The spores remain in the soil for several years and inoculum buildup across seasons exacerbates its impact on production systems [21]. The severity of symptoms, based on the proportion of affected tissue, varies and has been categorized into five classes, from two healthy kernels in a normal pod to two completely smutted kernels in a deformed pod [19,22]. Various strategies have been explored to manage peanut smut with poor to moderate and variable results, including the use of fungicides [9,17,19]. Within the latter, mixture formulations of cyproconazole and azoxystrobin led to only about a 50% reduction of incidence in field experiments with a disease pressure of 4400 teliospores g^−1^ of soil [23]. Among the available approaches, incorporating host resistance into breeding programs stands out as the most cost-effective and environmentally sustainable solution [24,25,26]. Although peanut smut remains geographically restricted and has not been reported in cultivated peanuts outside of Argentina, its rapid spread and the lack of effective control strategies have raised significant concerns among peanut research and production communities worldwide [23,27,28].

*Thecaphora frezzii* belongs to the phylum Basidiomycota, subphylum Ustilagomycotina, class Ustilaginomycetes, order Ustilaginales, and family Glomosporiaceae [29]. This fungus shares characteristics with other ‘true smut fungi’ within the order Ustilaginales that affect major agricultural crops such as wheat, maize, and barley, particularly in its biotrophic lifestyle, teliospore formation, and infection strategy. However, despite these similarities, *Thecaphora frezzii* exhibits a unique pathogenicity for peanuts, differing from cereal-infecting smuts that primarily colonize floral tissues, inflorescences, or developing grains. Instead, *T. frezzii* invades through the peg and colonizes the developing fruit, following a distinct infection route and tissue specificity [30,31].

In general, smut fungi share a consistent life cycle characterized by a parasitic dikaryotic phase and a saprophytic haploid phase [32]. The pathogenic process begins when two compatible haploid sporidia mate on the host surface, forming a dikaryotic mycelium capable of penetrating the host’s cuticle. This mycelium primarily parasitizes living cells in the vascular tissue, growing both intercellularly and intracellularly while extracting nutrients through a biotrophic interface [30]. In most cases, smut fungi infect their hosts at the seedling stage and spread systemically without causing visible symptoms initially. Observable symptoms, such as the restriction of inflorescences and the replacement of floral tissues with sori containing thick-walled black teliospores, typically emerge during the later stages of infection. These teliospores are adapted for long-term survival under harsh conditions and germinate under favorable conditions, producing promycelia that generate haploid basidiospores following meiosis. Typical symptoms of smut-infected crop plants are the formation of melanized, dark-pigmented teliospores in inflorescences, seeds, or leaves as well as leaf tissue rupture and stunted growth [29].

However, several smut fungi deviate from this pattern according to their life cycle, the plant organ they colonize, and the reproductive biology of the host plant [30]. While some smuts only invade the tissues around the initial penetration site, others colonize multiple plant organs resulting in spore formation distal from the original infection site. They show different directional growth in planta, varied in the induction of spore formation, and the plant endogenous signals required [30,31]. For instance, *Thecaphora solani* (Thirum. & M.J. O’Brien) Mordue, induces galls in subterranean stems and tubers [33] while *T. thlaspeos* (Beck) Vánky has teliosporogenesis in seeds of the Brassicaceae family [34]. Another example is *Ustilago maydis* (DC.) Corda infects all aerial parts of its host and induces localized smut galls within two weeks [35], while *Ustilago esculenta* P. Henn. causes smut galls in the stems of its host, *Zizania latifolia* Turcz., but exhibits a longer infection cycle [36]. In all cases, karyogamy occurs late in the infection cycle, producing diploid teliospores that form in inflorescent tissues or galls.

The reproductive biology of peanuts exhibits a distinctive mechanism in which flowering occurs above ground, while fruit development takes place subterraneously [1]. This process is mediated by the peg (gynophore), a specialized structure that exhibits positive geotropism, originating from a meristematic zone at the base of the ovary [37,38]. The gynophore elongates, transporting the fertilized ovary into the soil, typically at depths of 1–10 cm, where fruit development continues while maintaining vascular connectivity to the parent plant [39,40]. Despite advancements in characterizing this disease and how it affects peanuts [8,11,12,39,40], substantial gaps remain in our understanding of infection pathways, anatomical responses, and mechanisms of colonization and sporulation at the cellular and tissue levels. Given the peanut’s unique reproductive strategy, a deeper understanding of the histopathology of peanut smut is essential for elucidating disease development and advancing effective management strategies.

This study aims to comprehensively investigate the pathosystem and histopathology of peanut smut disease caused by *Thecaphora frezzii* in the susceptible cv. Granoleico peanut (Reg. No. 7907, El Carmen Nursery, Argentina) [16,41,42], focusing on its biotrophic cycle within the host. Given that the development of peanut reproductive structures follows well-defined phenological stages [43], understanding the timing and progression of fungal invasion in relation to these stages is essential. Specifically, the objectives are to: (i) trace the entry and progression of fungal hyphae through different peanut organs and tissues, detailing their proliferation within the peg, fruit, and seeds; (ii) describe the host tissues involved and the histological changes occurring in each organ and tissue at different developmental stages during fungal colonization; and (iii) determine the specific location and timing of teliospore formation in relation to the developmental stages of the peanut seed and fruit. To support these objectives, a detailed anatomical and histological analysis was conducted on non-invaded tissues of the organs that are later colonized by the fungus. A thorough examination of *T. frezzii* invasion, growth, and sporulation within host tissues will provide critical insights into its life cycle and the histological modifications induced in a susceptible peanut genotype during subterranean fruit development and disease progression.

## 2. Results

To systematically describe the colonization dynamics of *T. frezzii* in peanut tissues, the results are organized into sections that follow the sequential stages of fungal invasion, across different organs: the peg, the pod, and the seed. First, the phenological stages of fruit and seed development are described in relation to infection. The second section focuses on the biotrophic phase of the pathogen, covering its progression from initial penetration to sporulation. Finally, the third section presents the morpho-anatomical characteristics of the hyphae as they progress through the peanut tissues.

### 2.1. Developmental Stages of Peanut Fruit and Seed Associated with T. frezzii Colonization

The reproductive development of peanut is divided into nine distinct stages: R1 (beginning bloom), R2 (beginning peg), R3 (beginning pod), R4 (full pod), R5 (beginning seed), R6 (full seed), R7 (beginning maturity), R8 (harvest maturity), and R9 (over-mature pod).

Stage R2 (beginning peg) is characterized by the formation of the peg (gynophore) following fertilization (Figure 1A,B). This stage presents an aerial phase (R2-aerial peg stage) followed by the penetration of the peg into the soil (R2-subterranean peg stage), where only a change of coloration occurs. There are no visible external symptoms of the disease in these stages.

Boote’s R3 (beginning pod) stage marks the beginning of the development of the fruit and the seed (Figure 1A,B). At this stage, peg elongation ceases and the tip (with the young fruit) becomes curved and thickens to double the peg’s diameter, taking on the shape of a small shoe. At the end of this stage (named R3-late pod in this paper), the fruit wall has completed the formation of a mechanical layer, and the embryo has reached the torpedo stage. No external changes by *T. frezzii* are visible in the pegs or early stages of young pods (R3-beginning peg). The first noticeable morphological symptoms appear in the R3-medium pod to R3-late pod stages when the typical shoe shape of the pods begins to deform. In some cases, one of the seeds may abort, causing the fruit to adopt a spherical shape.

The severity of these malformations varies and progressively worsens as the seeds develop (R4 to R7 stages). The fruit reaches its final, cultivar-specific size at the R4 full pod stage. At this stage, fruits affected by *T. frezzii* show hypertrophy, the affected fruits become deformed or show an increase in diameter (Figure 1A,B). By the R5-beginning seed stage and in subsequent growth stages, the fruits exhibit hypertrophies that are visible to the naked eye (Figure 1A). The pods will show varying degrees of hypertrophy and a spongy consistency. The seeds are fully or partially transformed into carbonaceous masses of teliospores (Figure 1C), with one or both seeds potentially becoming infested.

### 2.2. Anatomy of the Biotrophic Stage of Infection of T. frezzii

#### 2.2.1. Fungal Colonization in the Peg (R2-Aerial Peg Stage to R3-Beginning Pod Stage)

The biotrophic phase of the smut *T. frezzii* consists of multiple stages that unfold sequentially as the peanut advances through its developmental phases, which have been previously characterized. Hyphae were not detected in the peg tissues at the R2-aerial stage. Their presence was first observed during the R2-subterranean peg and R3-beginning pod stages, when the gynophore, along with the young pod at its apical region, comes into contact with the soil and the teliospores present (Figure 2A,B).

Anatomy of the peg: The structure of the peg corresponds to the typical anatomy of the herbaceous dicot stem, consisting of a parenchymatous cortex, a vascular cylinder formed by an eustele with open collateral bundles, and a parenchymatous medulla (Figure 2A,C). Once buried, the epidermis develops unicellular, root-hair-like trichomes, and a subepidermal phellogen differentiates, forming lenticels (Figure 2C,D). The cortex of the peg consists of 8–10 compact layers of isodiametric parenchymatic cells with thin cell walls and few starch grains (Figure 2C). The vascular cylinder of the peg is composed of 11–13 open collateral bundles with an interfascicular cambium (Figure 2C). Each vascular bundle contains a cap of periphloematic fibers. At the innermost end of each bundle is a cluster of three to six tannin cells. The entire vascular bundle is surrounded by a sheath of parenchyma cells (Figure 3A,B). The pith of the peg consists of parenchyma with large polygonal cells rich in starch grains and intercellular spaces (Figure 2C).

Colonization: The infective hyphae of *T. frezzii* in the soil penetrate the peg through the epidermis and traverse cells intracellularly (Figure 2D). No penetration was observed via hairs, stomata, or lenticels.

In the epidermal and subepidermal cells invaded by the hyphae, condensed cytoplasm is observed, accompanied by an increase in the thickness of their cell walls (Figure 2D–F). The cells of the phellogen and the parenchymatic cells of the cortex adjacent to those invaded by hyphae divide actively (Figure 2D–G). Cytokinesis occurs in a periclinal manner, following the direction of the hyphae, resulting in the formation of micro-nodules with cells arranged concentrically. No increase in cell size is observed in the derived cells. The thickening of cell walls in the cells of the center of micro-nodules is associated with callose deposition and the cellular deposits exhibit a weak reaction to histochemical tests for phenolic compounds (Figure 2G,H and Appendix A).

A single peg may contain one or multiple micro-nodules, each corresponding to a hyphal entry point (Appendix A). These micro-nodules do not cause any visible external changes to the peg and do not contribute to further tumor development.

Once the hyphae enter and micro-nodules form in the subepidermal region, the hyphae continue their endophytic growth predominantly in a horizontal direction, always intracellularly, traversing the cortical parenchyma cells toward the central cylinder of the peg (Figure 3A–C). In this region, the thickening of the cell walls traversed by the hyphae is still observed.

The hyphae reach the parenchyma sheath surrounding the vascular bundles (Figure 3B–E). From there, they follow a descending path through the interfascicular parenchyma, as well as the fascicular and interfascicular cambium, and the xylem and phloem parenchyma. Hyphae neither invade the pith (Figure 3B,C) nor reinvade the cortical parenchyma as they descend towards the fruit (Figure 3F,G).

In longitudinal sections, the micro-nodules become elongated as the hyphae move from the epidermis to the central cylinder and then vertically to the developing fruit (Figure 3F,G and Appendix A). Periclinal divisions occur in peripheral cells along the hyphal path (Figure 3F,G).

The hyphae, which descend from the peg, follow the vascular bundles vertically through the cambium or parenchyma cells; they do not penetrate the conducting cells of both vascular tissues (Figure 3H,I). In this vertical trajectory, micro-nodules do not form, although some cell divisions may occur in the cells through which the hypha passes (Figure 3H).

#### 2.2.2. Fungal Colonization of the Pod (R3-Beginning to R3-Medium Pod Stages)

When the hyphae reach the pericarp, the fruits are in the R3-development stage, which may occur during the R3-beginning or R3-medium pod stages (Figure 1 and Figure 4A–C). Occasionally, hyphal penetration occurs at the fruit’s apical portion, traversing the mesocarp to reach the vascular bundles. In this area, no increase in cell divisions or wall thickening is observed (Figure 4B,C).

Anatomy of the pod: At the R3-medium pod stage, the pericarp is composed of three parenchymatous layers: a thin exocarp, a mesocarp, and an endocarp. A highly branched network of collateral vascular bundles separates the mesocarp from the endocarp (Figure 4A–C). The endocarp is the most developed layer, comprising multiple layers of small, compact parenchyma cells. A meristematic layer originates from the innermost mesocarp layers, forming toward the interior of the vascular bundles (Figure 4D,F–I). This meristematic barrier comprises 3 to 5 layers of small, elongated cells with thin walls and elongated nuclei, arranged in different planes perpendicular to each other. As a result of this cross-plane arrangement, some cells appear elongated, while others appear smaller in section (Figure 4D,F,G). This meristematic barrier also forms between the peg and the basal portion of the fruit, where there are no vascular bundles (Figure 4A).

Colonization: The hyphae, which have entered through the vascular bundles of the peg, proliferate in the parenchyma of the mesocarp, always remaining close to the phloem (Figure 4D,E and Appendix A). From there, the hyphae penetrate the vascular bundles and the meristematic barrier and continue their invasion through the endocarp in the direction of the seeds (Figure 4F–I).

As the fruit grows (from the R3-medium to R3-late pod stage), the hyphae that previously traversed the mesocarp penetrate the meristematic barrier, invading the endocarp and progressing toward the seed (Figure 4D–I). None of the parenchyma cells in the different pericarp layers show any reaction to the presence of hyphae, except for occasional cell divisions. Their nuclei appear normal, with no nodule formation or cell wall thickening (Figure 4E,G,H,J). Micro-nodules in the outer mesocarp were observed only in one fruit where the apical seed had aborted, causing the fruit to become spherical. These micro-nodules had the same structure as previously described (Figure 4I).

Hyphae were not found to cross the space between the endocarp and the seed coat (Figure 4J).

#### 2.2.3. Seed Colonization (R3-Medium Pod to R3-Late Pod Stages)

Anatomy of the seed: At the R3-medium pod stage, the seed is characterized by a thick funiculus where the vascular supply entering the seed coat from the pericarp splits into numerous small bundles rich in procambium. These bundles branch extensively throughout the testa of the seed coat (Figure 5A–D). The testa consists of uniseriate outer and inner epidermis layers and a homogeneous, compact mesophyll made up of 10–15 layers of small, isodiametric parenchymatic cells. The tegmen is discontinuous, in the chalazal region, it comprises 3–5 layers of large, polygonal parenchymatic cells without intercellular spaces and a uniseriate inner epidermis with quadrangular cells, dense cytoplasm, and a cuticle (Figure 5D). On the seed flanks, the tegmen are reduced to 1–2 layers, corresponding to the parenchymatic layer and the inner epidermis. At the micropyle, the tegmen is often interrupted (Figure 5D,G,H). The inner epidermis of the inner integument with its dense cytoplasmic cells constitutes the integumentary tapetum. This layer is separated from the embryo sac by a cuticle. The embryo is at the globular to heart-shaped stage and has a prominent suspensor (Figure 5B). The endosperm remains nuclear, consisting of a coenocyte of cytoplasm and nuclei located peripherally in the embryo sac, the central region of which is occupied by a large vacuole (Figure 5A,B,D,G,H).

Colonization: R3-medium pod is the earliest developmental stage in which hyphae colonize the seed. From the endocarp, the only way the hyphae penetrate the seed is through the thick funiculus (Figure 5A–C). Hyphal invasion reaches the seed coat and mainly colonizes the testa, where it is widely ramified; except for conducting elements of the xylem, and is seen in practically every cell of the testa, including epidermal cells (Figure 5A–F,I–K).

Hyphae continue to grow in the few tegmen layers (Figure 5G,H). From there, they penetrate the cytoplasm of the nuclear endosperm, where their route ends, and they cease to proliferate. The penetration of the hyphae into the endosperm is rare in relation to the abundance of the hyphae that colonize the seed coat. Hyphal invasion is limited to a thin cytoplasmic zone of endosperm; hyphal invasion of a central zone occupied by a large vacuole does not continue. In other words, the invasion of hyphae into the central cavity of the liquid endosperm has never been observed. Neither the embryo nor the suspensor is invaded by the hyphae (Figure 5B and Figure 6A,B).

#### 2.2.4. The Sporulation Site (R4 Stage—Full Pod Stage and Subsequent Stages) 

Anatomy of the pod and seed: As the fruit reaches its maximum size at the R4-full pod stage and progresses to the subsequent stages, mechanical tissues develop in the pericarp. In the exocarp, the periderm is formed at the same time as the meristematic barrier cells are transformed into sclereids. These cells are elongated, with a narrow lumen and very thick lignified walls, organized in layers arranged in crossed planes, forming a truly mechanical barrier. In the pericarp, this sclerenchymatous barrier forms arches with Y-shaped projections, in the concavity of which the vascular bundles are located. The mesocarp does not increase in size, remaining outside the barrier, and the endocarp is inwardly delimited and is the thickest layer at the R4 full pod stage.

Simultaneously with the increase in fruit size, there is a rapid increase in seed size. The increase in seed size is mainly due to an increase in the volume of the endosperm. At these stages the endosperm consists of a thin peripheral layer of free nuclei embedded in a dense cytoplasm, but without significant organelles, and a huge vacuole-like compartment that occupies virtually the entire cavity of the embryo sac (Figure 6A,B and Appendix A). To the naked eye, this is the ‘liquid’ endosperm. Cellularization of the endosperm occurs only in the chalazal region, with 3–5 cell layers. As embryogenesis progresses to stages corresponding to R4–R5, the embryo adopts a torpedo shape. During this phase, the suspensor maintains its triangular structure, characterized by large cells.

Colonization: At this stage, hyphae continue to proliferate in the seed coat. This results in intense hyperplasia with an increase in the number of cell layers, particularly in the region of the seed coat close to the funiculus. There is also increased vascularisation in this area, with extensive branching of bundles, together with a significant increase in the number of parenchyma cells associated with the vascular tissues (Figure 6A,B and Appendix A). The hyphae do not colonize the embryo or the suspensor (Figure 6A–D and Appendix A).

Sporulation: The hyphae initiate the process of teliosporogenesis in the area of the seed coat close to the embryo (Figure 6A,B). The cells of the invaded testa hypertrophy, increasing significantly in size as teliosporogenesis occurs (Figure 6C–E). Eventually, as the embryo enlarges, the hyphae also colonize the thin endosperm, in the cellularized region, but they do not form teliospores there. The cells of the seed coat rupture, leaving their remnants among the teliospores (Figure 6F,G). Ripe teliospores are enclosed by the outer epidermis of the testa, which increases the thickness of the cell walls as the development of the pericarp continues (Figure 6C,D).

The carbonaceous mass of teliospores completely replaces the seed coat and is visible to the naked eye in fruits at R6 and the following stages. At first sight, even under magnification, it appears that the cotyledons were transformed into a carbonaceous mass replacing them (Figure 1C). However, histological analysis showed that the embryo is never invaded by hyphae and that it atrophies while the seed coat hypertrophies and is then completely replaced by teliospores (Figure 6B–G and Appendix A). The hyphae running along the pericarp did not form teliospores in any of the fruits examined in this study (Figure 5D).

The teliospores are permanently agglutinated into firm groups of 2 to 7-spored balls. Mature teliospores have thick walls and feature an irregularly verrucose to spiny surface ornamentation (Figure 6E–I).

##### 2.3. Hyphal Characteristics

Significant structural transformations occur throughout the stages of the infection pathway, from the initial penetration of dikaryotic hyphae into the host plant to the site of sporulation. During the initial phase, as the hyphae traverse the superficial layers of pegs, including the epidermis and phellogen, they consist predominantly of elongated and cylindrical cells (Figure 2D). Within the horizontal progression through the cortical parenchyma of the peg during the early phase of micro-nodule formation, the hyphae exhibit a branched, Y-shaped morphology and predominantly follow a linear trajectory (Figure 7A–D). Two distinct hyphal types are observed in this area, differentiated by their diameter. Thick hyphae range between 1.48–3.01 µm (SD: 0.41, X: 2.40) (Figure 7A–G), while thin hyphae, identifiable via SEM, measure between 0.31–0.53 µm (SD: 0.07, X: 0.43) (Figure 7A,D,E).

During the downward progression through the vascular bundles of the peg, the hyphae first traverse the mesocarp and cross the meristematic barrier into the endocarp. Numerous branch primordia appear as protrusions on the surface of the main hypha, and their abundance and close spacing give the hypha a nodular appearance (Figure 4B,C,E and Figure 8A–E). The nodular hyphae are septate, with Y-shaped septa, and have a thickened structure, ranging from 3.47–4.47 µm (SD: 0.37, X: 3.99) in diameter (Figure 8A–D). These primordia do not elongate or further develop, meaning that no additional branching is observed in this section of the path. In the hyphae crossing the cortical parenchyma towards the vascular bundles, no branch primordia were observed (Figure 7A–G). The seed coat also serves as a favorable site for the observation of hyphal branching and septa between cells, as this area of colonization represents the peak of mycelial proliferation, occurring just before sporulation (Figure 8E–K).

Throughout this biotrophic phase, the hyphae always grow intracellularly within the plant tissues, and never invade the intercellular spaces (Figure 2D, Figure 3H,I, Figure 4C,E,J, Figure 5E, Figure 7A–G and Figure 8B,H–J). The movement of *T. frezzii* hyphae to new *Arachis* cells takes place via primary pit field sites, where plasmodesmata accumulate. This cell-to-cell progression induces the formation of a specialized hyphal structure known as a transpressorium (Figure 8F–I). Upon encountering a primary pit field, the hyphae swell and subsequently constrict as they traverse the intercellular communication pathway).

In the final stage of colonization, branch primordia develop within the seed coat, leading to intense branching of the mycelium previous to teliosporogenesis (Figure 8J,K). Once the hypha invades the seed coat, its diameter of 1.66–2.74 µm (SD: 0.37, X: 2.18) gradually decreases as it prepares for the sporulation process. The detailed visualization of the interaction between the fungal hyphae and the peanut cells, with their branching and transpressoria, is best seen in 3D images and videos (Figure 8H,I, Appendix A).

## 3. Discussion

It has been well documented in recent reviews that the biological cycle of smut fungi is highly conserved and shares many common aspects [31,32]. This is particularly evident in the saprophytic phase, whereas the biotrophic phase exhibits the greatest variability, with specific characteristics related to the intrinsic biology of the fungus, the host, and their interactions.

In this study, we provide a detailed description of the biotrophic cycle of *Thecaphora frezzii* in peanut (*Arachis hypogaea*) cv. Granoleico. To facilitate a better understanding of the action of *T. frezzii* in peanuts, Figure 9 provides a schematic summary of the biotrophic phase of the fungus across key developmental stages of the crop. The essential details are discussed below.

This cycle is unique in that it occurs entirely in subterranean organs, from entry through the peg structures to teliosporogenesis inside the buried fruits. This characteristic is exclusive to *T. frezzii*, as the only other smut fungus with a fully subterranean biotrophic cycle is *T. solani*. However, in *T. solani*, sporulation occurs in galls located on the lower stems, stolons, and tubers of potato plants (*Solanum tuberosum* L.), rather than in fruits [33,44]. On the other hand, in smut fungi where teliosporogenesis has been described in fruits and/or seeds, these reproductive structures always develop in aerial organs [30,34,35]. 

### 3.1. Hyphae Entry and Colonization of Peanut Pegs by T. frezzii

Previous studies have shown that *T. frezzii* preferentially targets and penetrates pegs and young fruits [12], where infection is likely initiated by peg exudates inducing teliospore germination, followed by the mating of compatible hyphae to produce the infective hypha [8,9,13,14,45]. However, they did not distinguish between fungal entry at the peg versus the ovary tip. A comprehensive analysis of the results here obtained indicates that the biotrophic phase of *T. frezzii* follows a well-defined pathway, entering through the distal portion of the peg, rather than the ovary at its tip, and progressing toward the peanut fruit and seeds, without exhibiting systemic colonization. Fungal entry occurs specifically during the R2-subterranean phase, marking the precise developmental stage of penetration in cv. Granoleico peanuts [Figure 9(1)].

Pegs arise from an intercalary meristem at the ovary base, elongating with positive gravitropism to bury the fertilized ovary and form an underground pod [38,46]. During this phase, embryo and endosperm development halts temporarily [40,47]. Intense cell division in the distal peg causes subepidermal and interfascicular cells to dedifferentiate, forming lateral meristems, key to secondary growth. These highly active zones act as strong metabolic sinks [48,49]. We propose that peg growth, as a major sink for photosynthates, may enable initial fungal hyphae colonization. Previous reports describe fungal entry through leaves, roots, staminal filaments, stigmas, and developing seeds across different smut-causing genera, including *Thecaphora, Sporisorium*, and *Ustilago* [30,31,50,51]. Thus, the entry of *T. frezzii* hyphae through the peg represents a novel infection route among smut fungi.

### 3.2. First Challenges: Overcoming Peanut Plant Defenses

Following hyphal entry through the epidermis, *T. frezzii* exhibits a straight growth toward vascular bundles, a feature commonly shared with many other smut fungi [30,31,35,44]. In *T. frezzii*, this initial linear growth—referred to here as the horizontal phase—occurs intracellularly without inducing exo-morphological changes in the peg. The ability of hyphae to penetrate without inducing externally visible infection symptoms or causing drastic host cell damage is a common characteristic among smut fungi, particularly in those with systemic infections [30].

Histologically, after penetrating the epidermal cells, *T. frezzii* hyphae encounter a meristematic layer, the phellogen, as well as cortical parenchyma. These cells exhibit two distinct responses to fungal presence: cell wall thickening and the formation of localized micro-nodules [Figure 9, inset 1]. Cell wall thickening through callose and lignin deposition is a well-documented defense against pathogenic fungi, including smuts [30,32]. In sugarcane smut caused by *Sporisorium scitamineum* (Syd.) M. Piepenbr., M. Stoll & Oberw., these compounds contribute to the formation of a defensive barrier in resistant cultivars [52,53]. Similarly, lignin accumulates in maize in response to *Ustilago maydis*-induced tumors [54,55], while callose is observed in barley infected by *Ustilago hordei* (Pers.) Lagerh. [56]. Notably, cv. Granoleico is among the most susceptible to peanut smut [41]. In the *T. frezzii*-peanut cv. Granoleico pathosystem here analyzed, this response is insufficient to halt fungal progression, as hyphae rapidly grow and overcome these initial barriers.

The second histological response in peanut pegs, micro-nodule formation at the hyphae entrance, is novel for any smuts so far [33,34,35,55,57,58,59]. Initially, infected cells remain metabolically active and capable of mitosis. Micro-nodule cells exhibit two division traits: (i) radially oriented planes, always perpendicular to hyphal growth, and (ii) no expansion, so daughter cells do not grow. As *T. frezzii* colonizes new cells, micro-nodule division quickly stops. This results in localized hyperplasia—an increased cell number—without altering the peg’s macroscopic structure. Notably, no hypertrophy (pathological cell enlargement) occurs in these nodules, aligning with reports of infected pegs appearing macroscopically ‘asymptomatic’ [8]. 

Micro-nodule formation represents the first anatomically conspicuous sign of the peanut-*T. frezzii* interaction. This response may be somewhat homologous to tumor formation in other smut fungi, though with key differences. In previously studied pathosystems, hyphal-induced cellular proliferation typically occurs in reproductive or vegetative organs at more advanced biotrophic phases [55,57,58,59]. Examples include: *U. maydis*, which induces tumor formation in maize [35]; *T. solani*, which causes potato smut [33]; *T. thlaspeos*, associated with neoplastic growths in *Brassicaceae* spp. [34]. However, no previous studies have reported an early infection response involving micro-nodule formation similar to what is observed in the *T. frezzii*–peanut pathosystem.

### 3.3. Second Challenge: Acquiring Nutrients from the Host During Colonization

Once hyphae reach the central cylinder, they undergo a sharp (ca.~90°) directional shift, invading vascular bundles and following the phloem flow towards the fruit and seed (Figure 9, insets 2–4). The phloem is a tissue that connects source and sink organs, transporting amino acids, RNAs, organic acids, vitamins, and soluble carbohydrates [60,61]. The directional growth of *T. frezzii* hyphae towards and through the vasculature after initial penetration may be related to nutrient gradient sensing, allowing the fungus to position itself close to accessible host nutrients, as occurred in other smut fungi pathosystems [30,34]. In *A. hypogaea*, the peg connects autotrophic aerial organs (sources) with the underground fruit and seeds (strong sinks) [38,39], creating a nutrient gradient that may direct hyphal movement. Similar guidance mechanisms based on soluble hexose concentration differentials have been proposed for fungal colonization [32].

Despite previous reports suggesting that fruit colonization occurred via the phloem, the exact cell types invaded by the hyphae were not specified [12,13]. This study is the first to identify the specific vascular bundle cells utilized by *T. frezzii* to colonize the peg, fruit, and seed. Among the available cell types within the vascular bundles and interfascicular tissues, hyphae exclusively invade metabolically active, living cells, including phloem parenchyma, interfascicular and intrafascicular meristematic cells, xylem-associated parenchyma, and bundle sheath cells. These cell types were those preferred by the hyphae to establish the colonization pathway. In contrast, hyphae do not traverse pith parenchyma cells nor do they directly invade xylem vessels or sieve tubes at any point in their journey from the peg to the seed. While directional hyphal growth through vascular tissues is a known characteristic of smut fungi, prior studies have not identified the specific host cells involved [e.g., *Ustilago scitaminea* Sydow, Ann., *U. maydis, T. thlaspeos* [30,34,62]]. One of the few cases that describes the hyphal pathway associated with vascular bundles, particularly in phloem companion cells and xylem parenchyma, is *Sorosporium caledonicum* Pat., which infects *Heteropogon contortus* (L.) Beauv. This pattern is similar to what we observed in *T. frezzii*, where hyphae also exhibit a preference for these specific vascular-associated cell types [63].

Since *T. frezzii* is a biotrophic pathogen that requires living host cells for nutrient uptake, establishing a close interaction with the host while minimizing damage [29,31], its preference for active vascular bundle cells may reflect a strategy to sustain long-term colonization from the peg to the seed coat. From studies on other smut fungi, it is evident that an efficient nutrient supply from the host is essential for the massive fungal proliferation observed in tumor tissue [31,64]. Unlike investigations in *U. maydis*, which suggest a redirection of carbohydrates from photosynthetically active organs to infected tissues, where assimilates preferentially accumulate in tumor tissue [65], *T. frezzii* appears to follow the natural sugar gradient occurring in actively dividing tissues during peanut fruiting. This process may be further enhanced by inducing localized hyperplasia and hypertrophy in the final destination organs, i.e., the fruit and seed, as observed in this study (Figure 9, insets 4–5).

### 3.4. Fruit Colonization

Within the fruit, *T. frezzii* hyphae keep its growth closely associated with the vascular bundles of the mesocarp, bypass the meristematic barrier, and colonize the endocarp, consistently following the sugar transport pathway (Figure 9, inset 3).

Fruit colonization occurs during the R3 stages (R3-beginning and R3-medium) when a meristematic zone between the mesocarp and endocarp is still present. This region provides an easily penetrable pathway, as the sclereid barrier has not yet developed. Although several smut fungi, such as *U. maydis* or *T. thlaspeos*, are characterized by their ability to colonize kernels [34,35], the invasion of the fruit wall as part of the route toward the seeds as described here for *T. frezzii* has not been histologically documented in other dicot hosts affected by smut fungi.

Hyphae colonizing the endocarp far from the funiculus fail to progress their growth toward the seed, seemingly unable to grow in the space between the endocarp and seed coat. The absence of hyphae in the interstitial space between these layers may be explained by the strictly intracellular growth of *T. frezzii* during colonization. As a consequence, the funiculus arises as the only entry point for the hyphae into the seed through its vascular bundles.

This invasion of the fruit induces hyperplasia and hypertrophy in both the mesocarp and endocarp, leading to fruit deformation, which constitutes the only exomorphological symptom (paper in progress) cited as a diagnostic indicator of *T. frezzii* infection in the final stage of its biotrophic cycle in peanuts [8]. Although these anatomical changes resemble the gall formation process that precedes teliosporogenesis in other smut fungi (e.g., in *Thecaphora solani* [33] and in *Ustilago maydis* [54]), in peanuts, hyphae triggering hyperplasia and hypertrophy in the fruit do not initiate teliosporogenesis.

### 3.5. Seed Colonization

Developmentally, a seed comprises a diploid (2n) embryo and reserve tissues, including the triploid (3n) endosperm or nutrients stored in the diploid (2n) cotyledons, all enclosed by a maternally derived seed coat. This structure is connected to the ovary solely through the funiculus, which serves as the entry point for nutrients via the vascular bundles [66]. In peanuts, hyphal entry to the seed occurs between the R3-medium and R3-late pod stages via the funiculus, followed by extensive colonization of the seed coat tissues without penetrating the embryo (Figure 9, Insets 4–5). Once inside the seed coat, a second phase of massive hyperplasia and hypertrophy is triggered, at the time that occurs the teliosporogenesis (Figure 9, Inset 6). In Fabaceae, and particularly in *Arachis* L., the seed coat features an intricate, anastomosed network of vascular bundles, which remains confined to the seed coat and does not extend beyond it [47,67]. This characteristic helps explain the extensive proliferation and branching of hyphae observed in the peanut seed coat, which has also been reported in other smuts [68].

Upon reaching the testa and the thin tegmen, hyphae may penetrate the endosperm but only into the adjacent dense cytoplasmic layer containing non-cellularized nuclei. While some hyphae extend into the liquid-filled endosperm space, they do not proliferate, likely due to their exclusive intracellular growth. This restriction limits further colonization in these cell-free regions, marking the end of hyphal progression within the seed.

Photoassimilates, primarily sucrose, produced in the autotrophic tissues of the plant are transported through the phloem along the funiculus and delivered to the seed coat, which both protects and nourishes the embryo during development [69,70,71,72]. The embryo and endosperm remain apoplastically isolated from the maternal plant [70,71,72,73].

Our results demonstrate that *T. frezzii* hyphae are confined by the apoplastic barriers within the seed coat, preventing them from entering or colonizing the embryo. They also demonstrate that the seed coat, characterized by an extensive vascular network, rather than the embryo, serves as the most important site for intense hyphal proliferation during peanut smut infection.

In some smut fungi, the ovary or seeds serve as primary sites for tumor formation and teliospore production, disrupting seed development or leading to partial or complete replacement of seeds by teliospores (e.g., *Thecaphora amaranthicola* M. Piepenbr. [74], *T. thlaspeos* [34], *Ustilago maydis* [55], *U. tritici* (Pers.) Rostr. [75]). However, no detailed histological studies exist on fungal colonization of the different seed structures.

The detection of teliosporogenesis occurring exclusively in seed coat at the onset of accelerated embryo growth allowed a reinterpretation of the effects of *T. frezzii* on peanuts so far published [9,31]. Here we demonstrated that in severe infections, teliospores completely occupy the space where the embryo would have developed, enclosed by the external seed coat epidermis. Notably, the observation that the extensive hyphal colonization occurred in all cellular layers of the seed coat, except the outer epidermis offers an explanation of the origin of the in-fruit contention structure of teliospores at the end of the fungus cycle. The massive teliosporogenesis utilizes the nutrients of the seed coat, preventing embryo development beyond the torpedo stage, likely as a result of nutrient deprivation. Thus, embryonic tissues are not physically replaced by teliospores, as previously proposed for peanut smut [8,9,12,31].

Teliosporogenesis in the seed coat leads to the second major symptom of the disease—the formation of a dense, carbonaceous mass of teliospores occupying the embryonic space in a mature seed. Variability in disease severity observed in peanuts [9,22] may be linked to the timing of hyphal invasion and seed coat colonization. This timing may influence the onset of massive apoplastic nutrient translocation from the seed coat to the endosperm and embryo, and therefore, different degrees of embryo development and the number of teliospores in the fruits.

### 3.6. Hyphal and Teliospores Morphology

The presence of septate hyphae, their intracellular growth, and the overall structure of the teliospores of *T. frezzii* analyzed in this study align with previous descriptions for this species [19,31,76] and other smut fungi [77,78]. However, our findings reveal that during colonization, *T. frezzii* hyphae exhibit diverse morphologies and undergo structural modifications that vary depending on the infected tissue and the specific organs they colonize.

Two characteristics observed in this study are novel for the genus *Thecaphora:* branch primordia and transpressorium. Firstly, during vertical colonization, as the fungus spreads through the peg and fruit towards the seed coat, hyphal growth is predominantly linear, with numerous branch primordia. These structures appear as short branches that do not fully develop and possess a Y-shaped associated septum, resembling the clamp connections observed in *Ustilago maydis* hyphae [79,80,81,82].

In a previous study, hyphae forming short branches ending in a dilatation observed in the pericarp of peanut fruit were interpreted as haustoria [11]. However, haustoria are characteristic structure of rust fungi, which exhibit intercellular growth. These specialized organs develop within a living host cell as terminal branches of hyphae, likely facilitating the exchange of substances between the host and the fungus [83,84,85]. Our findings align with observations in *U. maydis* [35], where hyphae do not form haustoria, suggesting that signal exchange and nutrient absorption occur directly through the biotrophic interface.

Secondly, the novel structure described here for *T. frezzii* is the transpressorium, observed at sites where hyphae penetrate from cell to cell [86]. During intracellular migration, hyphae swell and subsequently constrict to traverse primary pit fields. The morphology of this structure is consistent with that described in the rice blast fungus *Magnaporthe oryzae* B.C. Couch [86,87]. The identification of the transpressorium in *T. frezzii* provides a better understanding of the fungal colonization mechanism of *Arachis hypogaea*. This structure may allow the pathogen invasion of neighbor cells while maintaining cell integrity during invasion, a key feature of its biotrophic phase. This discovery paves the way for future studies aimed at elucidating the mechanisms that enable the fungus to recognize intercellular junctions as passage sites between adjacent host cells.

## 4. Conclusions

In this study, through a detailed description of the biotrophic cycle of *Thecaphora frezzii* in peanut cv. Granoleico filled numerous gaps in the knowledge of this particular pathosystem.

We clearly identified the peg, rather than the ovary, as the entry organ for hyphae. Additionally, we established the complete intracellular pathway, identified the cell types used by the hyphae to progress in colonization, and pinpointed the specific tissues where sporulation occurs. The seed coat, of maternal origin, is the only site for the simultaneous development of hyperplasia, hypertrophy, and teliosporogenesis in peanut smut, rather than in the embryo tissues as previously considered. The symptom of abnormal and hyperplasic fruits depends on the occurrence of hyperplasia and hypertrophy in the pericarp, a mechanism that appears more or less independent of the teliosporogenesis in the seed coat.

We also recognized a precise timeline for fungal entry and progression, closely linked to fruit developmental stages, which is essential for successful colonization and sporulation. Several hypotheses were proposed to explain the selection of this pathway for the biotrophic cycle, though their validation requires further experimentation, some of which is currently underway.

Finally, we provide a general picture of the plant structural reactions during the fungus colonization. Studying these structural responses in phenotypically contrasting genotypes from peanut germplasm may provide insights into the mechanisms involved in disease susceptibility/resistance. Ongoing research aims to determine whether cell wall thickening and micro-nodule formation may actively restrict fungal growth or may serve as a passive barrier in smut-tolerant/resistant genotypes of peanuts.

## 5. Materials and Methods

### 5.1. Plant Material

The commercial cultivar *Granoleico* is one of the most susceptible to peanut smut [41]. Plants of this cultivar were obtained from El Carmen Nursery, located in General Cabrera, Córdoba, Argentina (32°49′40.8″ S, 63°52′14.0″ W). Selected plants for this study were obtained from a smut evaluation trial conducted during the 2021–22 and 2022–23 growing seasons. The trial consisted of single-row plots (2.5 m long) arranged in a randomized complete block design with two replicates, planted in soils containing an average of 12,000 *T. frezzii* teliospores per gram of soil. This inoculum density is three times higher than the average concentration (3000–4000 teliospores g^−1^) observed in naturally infested peanut fields within Argentina’s peanut-growing region [88]. Standard agronomic practices were applied, and plants were sampled at the end of the growing season (110–120 days after sowing) when smut incidence peaks [9].

For the present study, pegs and pods at developmental stages ranging from the beginning peg (R2) to full seed (R6), as defined by Boote [43], were collected and processed. Ten plants were selected, from which 20 pegs with fruits were analyzed for the R2 and R3 stages, while five samples per stage were processed for R4, R5, and R6. The samples were fixed in FAA [formaldehyde (37%), 70% ethanol (70%), glacial acetic acid (100%); final proportions: 5:90:5] in the field. This fixation method was consistently used for all subsequent microscopy techniques (LM, CLSM, and SEM). 

### 5.2. Light Microscopy (LM)

The infected material underwent histological dehydration using Biophur^®^ (Rosario, Argentina) histological dehydrants [89]. This process included five changes in pure dehydrants, each lasting 20 min, followed by two changes in a tertiary butanol series (8 h and 2 h, respectively). The samples were then infiltrated with two changes of pure Histowax (Biopack, Buenos Aires, Argentina) at 60 °C for 8 and 2 h, respectively [90]. Serial transverse and longitudinal sections, 10–12 µm thick, were obtained using a MICROM International GmbH (Walldorf, Germany) rotary microtome. The sections were mounted on slides using Haupt’s adhesive and dried for 48 h before staining [91]. The slides, containing sections from each developmental stage, were divided into three sets for observations under light microscopy (LM), scanning electron microscopy (SEM), and confocal laser scanning microscopy (CLSM).

For LM analysis, the sections were dewaxed with xylene using two changes of 10 min each. The slides with the sections were then rehydrated through a descending ethanol series until reaching 50% ethanol [90]. Staining was performed with 1% Safranin Biopack and 0.5% Astra blue Biopack (in 50% ethanol, for 15 min each) [92]. The sections were then dehydrated again through an ascending ethanol series and xylene before being mounted in synthetic Canada balsam Biopack [90]. Phenolic compounds were detected by treating the sections with 5% ferric chloride [93], and lignin was identified using 0.1% phloroglucinol Merck KGaA (Darmstadt, Germany) [90,93]. Observations and imaging were performed using a Leica DM LB2 microscope equipped with a Leica ICC50W digital camera (Leica Microsystems, Wetzlar, Germany).

### 5.3. Confocal Laser Scanning Microscopy (CLSM)

Paraffin-embedded material was sectioned at a thickness of 12 μm, deparaffinized, and rehydrated to 50% ethanol (according to the LM methodology). The sections were mounted in pure glycerol without staining for autofluorescence analysis. Additionally, they were stained with Calcofluor White M2R (Sigma-Aldrich, St. Louis, MI, USA), following the manufacturer’s instructions. For staining, one drop of the dye was mixed with one drop of 10% Potassium hydroxide, covered with a coverslip for one minute, and observed using CLSM with excitation/emission wavelengths of 425/475 nm [94].

Confocal images were acquired using a Stellaris 8 White Light Laser inverted confocal microscope (Leica Microsystems, Wetzlar, Germany) equipped with HC PL APO 10× dry (NA 0.4) and 63× oil immersion (NA 1.4) objectives. Excitation/emission wavelengths were set as follows: blue, 405/413–489 nm; green, 503/508–633 nm; red, 627/635–750 nm. Images and three-dimensional (3D) reconstructions were generated using Leica LAS X Stellaris Compass software (version 5.3.0; Leica Microsystems, Wetzlar, Germany). The LAS Navigator module (part of the LAS X platform; Leica Microsystems, Wetzlar, Germany) was used in conjunction with the 63× objective to view and merge tiles into a single composite image. Serial optical sections were taken in xyz mode to produce 3D views of the specimens, extending from the surface to the required depths. Z-stacks or parts of them were projected into 2D images. Depth-coded Z-stack images are also presented where the depth gradient follows the area of the spectral light (uppermost structures of the stack appear red, deep ones blue).

### 5.4. Scanning Electron Microscopy (SEM)

Paraffin-embedded materials (see technique in LM sections) were processed for SEM following the protocol described by Gonzalez et al. [95]. According to this protocol, 30-µm-thick serial sections were cut using a rotary microtome and mounted on glass slides with Haupt’s adhesive [91], then left to dry for 48 h. To accommodate the critical point drying process, the slides were pre-cut to a length of 4 cm to fit into the chamber of the critical point drying unit (Denton Desk II, Moorestown, NJ, USA). The sections were then deparaffinized in xylene (two changes of 10 min each), rehydrated through a graded ethanol series (100%, 96%, and 70%, for 5 min each), followed by an ascending acetone series (70%, 90%, and 100%, for 5 min each), and subsequently dried to the critical point using CO_2_. The specimens were then coated with a gold-palladium layer and examined using a Jeol LV 5800 SEM (Tokyo, Japan) at 20 kV at the Electron Microscopy Service of UNNE, Corrientes. 

### 5.5. Measurements

For hyphal thickness measurements in the different invaded organs—peg, pod, and seed—images of hyphae obtained from LM, SEM, and CLSM observations were analyzed (n = 10 sections). They were taken from the main hyphal portions, avoiding branching primordia or branching zones. Measurements were performed using Fiji software (version 2.16.0) [96].

All observations and photographic captures were performed manually, without the use of an automated system, ensuring that no accidental shots were taken.

## Figures and Tables

**Figure 1 plants-14-01083-f001:**
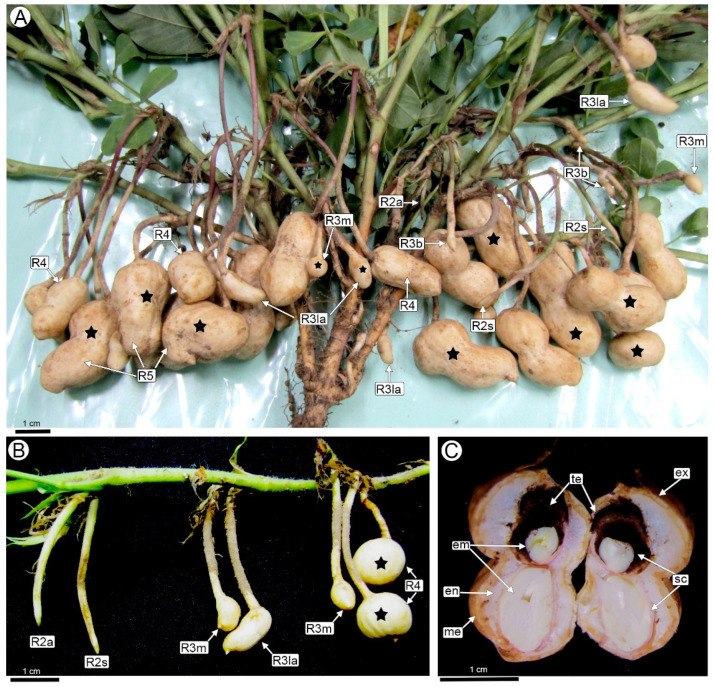
The growth stages of peanuts and the appearance of symptoms of smut caused by *T. frezzii.* (**A**) General appearance of a groundnut plant infected with *T. frezzii*, displaying fruits at different developmental stages, most of which exhibit hypertrophy (★); (**B**) Detail of a peanut branch showing pegs and pods at early stages of development; (**C**) Longitudinal section of a pod at the R6 (full seed) stage, showing the basal seed with a reduced embryo and seed coat invaded by teliospores, visible as a carbonaceous mass. The apical seed appears unaffected. Abbreviations: em: embryo, en: endocarp, ex: exocarp, me: mesocarp, R2a: R2-aerial peg, R2s: R2-subterranean peg, R3b: R3-beginning pod, R3m: R3-medium pod, R3la: R3-late pod, R4: R4-full pod, R5: R5-beginning seed, sc: seed coat, te: teliospores, ★: deformed pods.

**Figure 2 plants-14-01083-f002:**
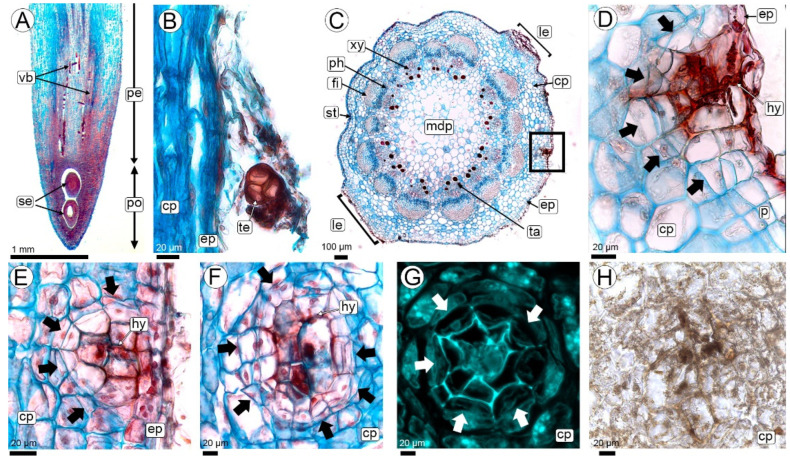
Penetration of *T. frezzii* hyphae on peanut pegs at R2-subterranean peg and R3-beginning peg stages. (**A**–**F**) Images obtained using light microscopy (LM), with staining by Safranin and Astra blue; (**A**,**B**,**E**) Longitudinal sections; (**A**,**B**) R2-subterranean peg; (**B**) Details of peg showing teliospore near epidermis; (**C**,**D**) Cross-sections; (**C**) Structural details of R3-beginning pod stage showing general structure and the hyphal entry zone (box); (**D**) Detail of hyphal entry region (corresponding to the square in photo (**C**)), showing cell divisions in the phellogen (arrows); (**E**) Micro-nodule formation via radial cell divisions (arrows) in the cortical parenchyma surrounding the hyphae; (**F**) Paradermal section of a micro-nodule, showing the arrangement of cells around the hyphae (arrows); (**G**) Confocal Laser Scanning Microscopy (CLSM) view of a micro-nodule, displaying thickened callose walls in the area around the hyphae, and thin walls in the radially arranged cells (arrows); (**H**) LM of micro-nodule stained with Ferric chloride, showing weak reaction to phenolic compounds deposits. Abbreviations: arrows: periclinal divisions, cp: cortical parenchyma, ep: epidermis, fi: fibrous cap, hy: hyphae, le: lenticel, mdp: medullary parenchyma, p: phellogen, pe: peg, ph: phloem, po: pod, se: seed, ta: tanniniferous cells, te: teliospore, vb: vascular bundle.

**Figure 3 plants-14-01083-f003:**
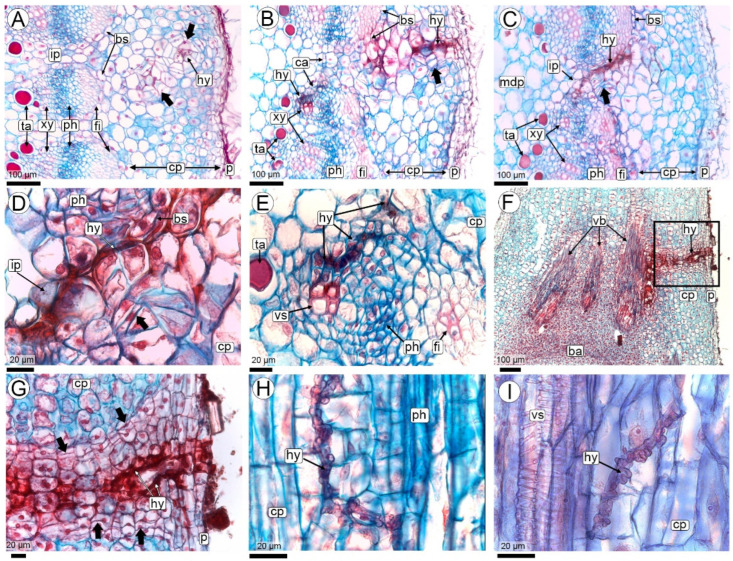
Colonization of pegs at R3 stage by *T. frezzii,* images obtained using light microscopy, stained with Safranin and Astra blue. (**A**–**E**) Cross-sections of pegs showing hyphal routes through the cortical parenchyma toward the vascular bundle regions; (**D**) Close-up of (**C**), highlighting hyphal pathways in the interfascicular cells and periclinal cell divisions at the periphery; (**E**–**I**) Longitudinal sections; (**E**) Route of the hyphae through vascular bundle cells; (**F**) Hyphal colonization in the cortical parenchyma of distal portion of peg (**G**) Detail of (**F**), showing hyphae and cell divisions in the cortical parenchyma; (**H**,**I**) Hyphae in the vascular bundle region. Abbreviations: arrows: periclinal divisions, ba: barrier, bs: bundle sheath, ca: cambium, cp: cortical parenchyma, en: endocarp, fi: fibrous cap, hy: hyphae, ip: interfascicular parenchyma, mdp: medullary parenchyma, me: mesocarp, p: phellogen, ph: phloem, ta: tanniniferous cells, vb: vascular bundles, xy: xylem.

**Figure 4 plants-14-01083-f004:**
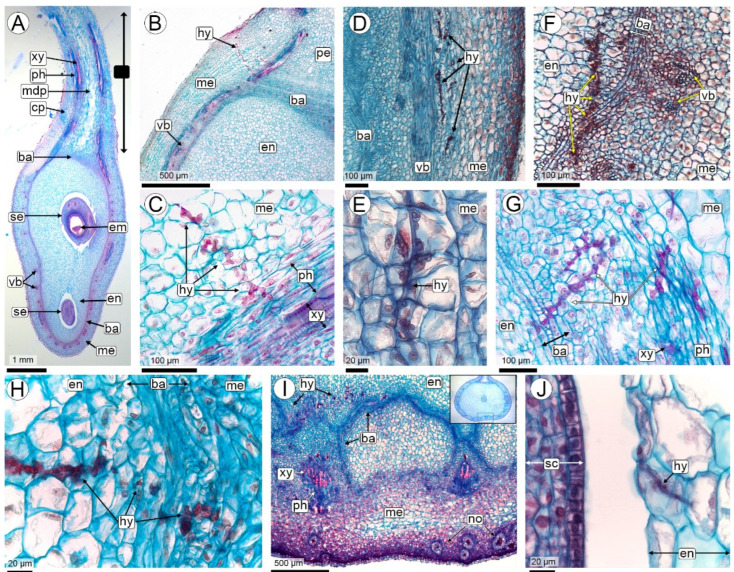
Pod colonization by *Thecaphora frezzii*. (**A**–**J**) Images were taken with light microscope and stained with Safranin and Astra blue. Longitudinal sections (**A**–**J**) Peg/pod at the R3-medium stage; (**B**) Peg-pod junction with visible hyphal penetration; (**C**) Close-up of (**B**), illustrating hyphal progression through the mesocarp parenchyma to the phloem; (**D**,**E**) Hyphae within the mesocarp; (**F**–**H**) Hyphae from mesocarp to endocarp, crossing the vascular bundles and meristematic barrier (arrows in (**G**) indicate cells of meristematic barrier oriented at 90° angles); (**I**) Pod at the R3-medium stage with a single seed (corresponding to the photo in the inset), showing the position of hyphae in the endocarp and subepidermal micro-nodule; (**J**) Endocarp near the seed coat, showing the presence of hyphae; note that they do not cross the space to invade the seed. Abbreviations: ba: barrier, cp: cortical parenchyma, em: embryo, en: endocarp, hy: hyphae, mdp: medullary parenchyma, me: mesocarp, no: micro-nodule, ph: phloem, sc: seed coat, se: seed, vb: vascular bundle, xy: xylem.

**Figure 5 plants-14-01083-f005:**
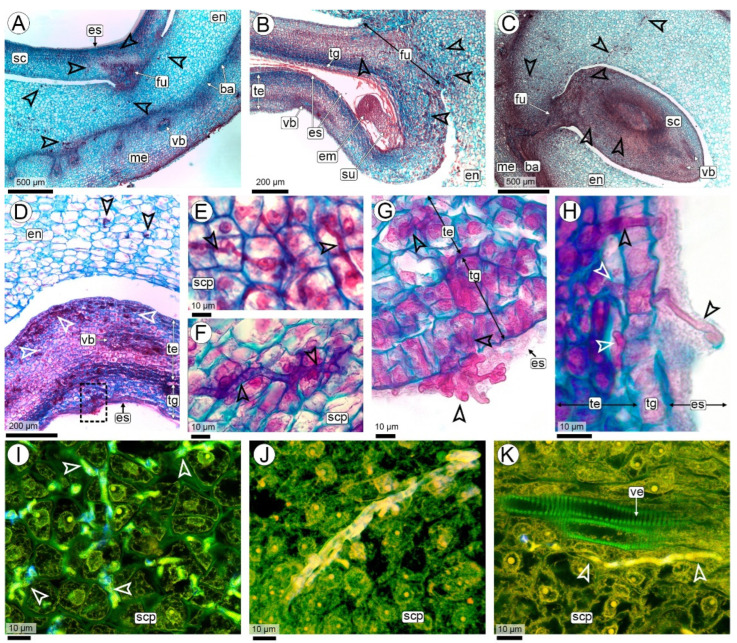
Seed coat colonization at R3-medium and R3-late pod stages. (**A**–**H**) Images taken with a light microscope and stained with Safranin and Astra blue. (**A**–**C**) Seed-pod attachment showing the path of hyphae from the endocarp through the funiculus to the seed coat; (**D**) Detail of the endocarp and seed coat invaded by hyphae; (**E**,**F**) Hyphae in the testa; (**G**) Chalazal region of the tegmen (corresponding to the boxed area in photo (**D**)) showing hyphae proliferating in the endosperm; (**H**) Lateral side of seed showing hyphae proliferating through tegmen and endosperm; (**I**–**K**) CLSM images of hyphae in the testa. Abbreviations: arrowheads (white or black): hyphae, ba: barrier, en: endocarp, es: endosperm, fu: funiculus, me: mesocarp, sc: seed coat, scp: seed coat parenchyma, te: testa, tg: tegmen, vb: vascular bundle, ve: vessels of xylem.

**Figure 6 plants-14-01083-f006:**
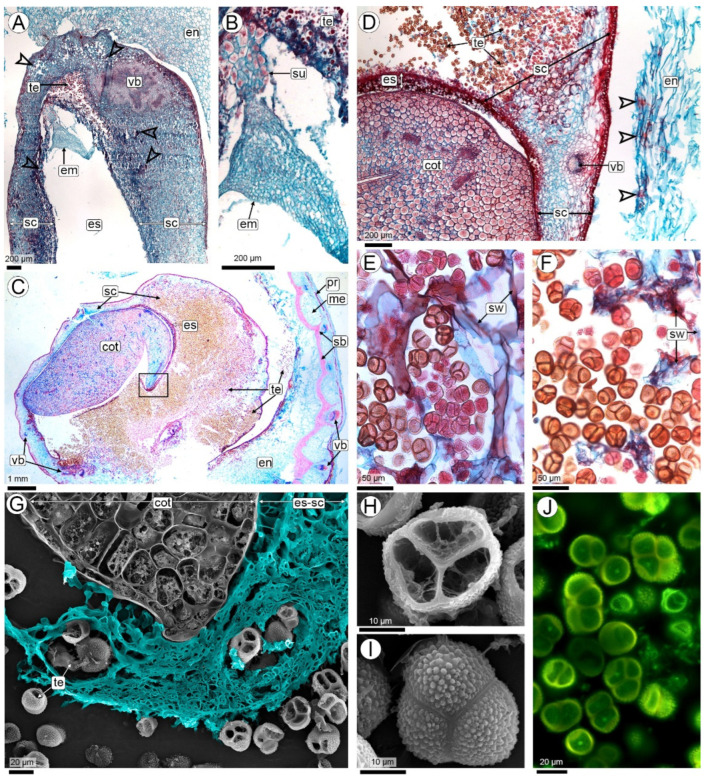
*Thecaphora frezzii* sporulation. (**A**–**F**) Images taken with an LM and stained with Safranin and Astra blue; (**G**,**J**) CLSM; (**H**,**I**) SEM. (**A**) Seed at the R5 pod stage; (**B**) Detail of the embryo in (**A**); (**C**,**D**) R6-full seed stage, (**C**) Hypertrophied testa cell containing young and mature teliospores; (**D**) Detail of a hypertrophied seed coat, healthy cotyledon, and pericarp; (**E**) Teliospores in hypertrophied cells; (**F**) Teliospores and cell lysis; (**G**) Enlarged section (corresponding to the boxed area in photo (**C**)) showing a cotyledon without hyphae and endosperm and testa completely occupied by hyphae (digitally colored) and teliospores; (**H**–**J**) Mature teliospores. Abbreviations: arrowheads: hyphae, cot: cotyledon, em: embryo, en: endocarp, es: endosperm, pr: periderm, sb: sclerenchymatous barrier, sc: seed coat, sw: seed coat walls, vb: vascular bundles, te: teliospores.

**Figure 7 plants-14-01083-f007:**
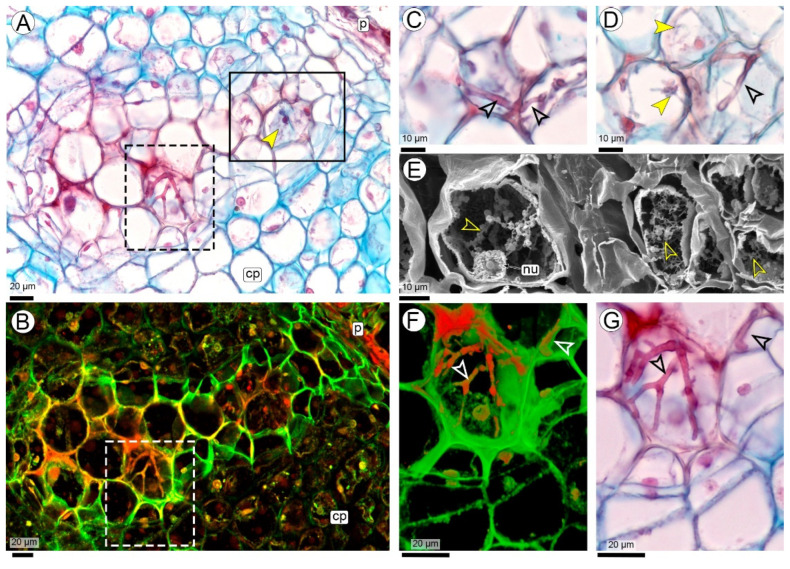
Characteristics of the hyphae of *T. frezzii* in the horizontal route at the entry zone of the peanut pegs at R3-beginning stage of development. (**A**,**C**,**D**,**G**) Images taken with a light microscope and stained with Safranin and Astra blue; (**B**,**F**) The same slide analyzed with CLSM; (**E**) SEM. (**A**,**B**) Cross section of the peg with micro-nodules. Note the thick, branched hyphae (boxed area with dashed lines) and the thin hyphae (boxed area with solid lines); (**C**,**D**) Details of thick and (**E**) thin hyphae. (**F**,**G**) Details of ramified hyphae corresponded to boxed areas with dashed lines at (**A**,**B**) photos. Abbreviations: black and white arrowheads: thick hyphae, yellow arrowheads: thin hyphae, cp: cortical parenchyma, nu: nucleus, p: phellogen.

**Figure 8 plants-14-01083-f008:**
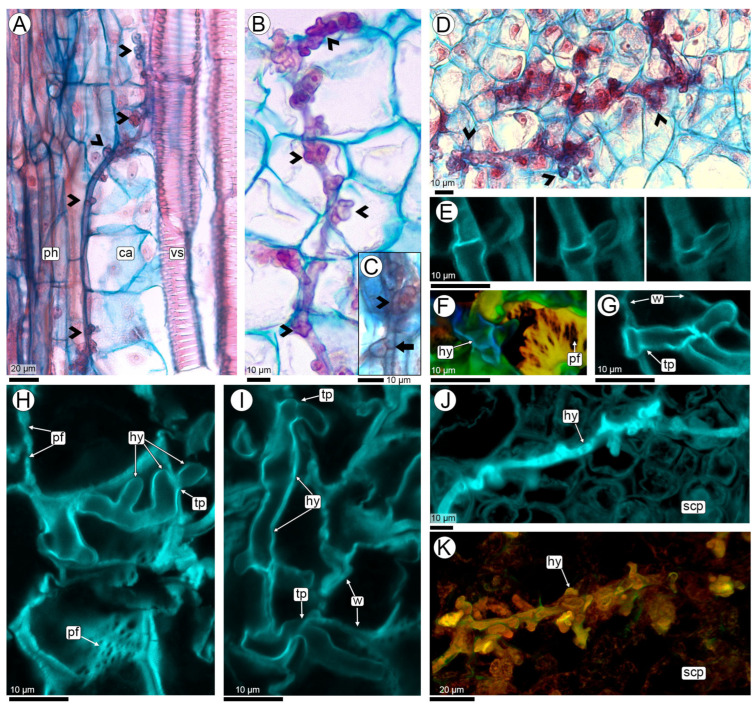
Characteristics of the hyphae in their vertical route. (**A**–**D**) Images taken with a light microscope, stained with Safranin and Astra Blue, showing longitudinal sections of tissues colonized by nodose hyphae with multiple branch primordia. (**A**) Transition zone between the peg and the pod, the hyphae are located in the cambium between the xylem and the phloem; (**B**) Mesocarp; (**C**) Close up of photo (**B**) showing a couple of branch primordia and Y-shaped septum (arrow); (**D**) Endocarp; (**E**–**K**) Images of seed coat with CLSM; (**E**) Sequence of three focal planes showing a hyphal branch in the seed coat; (**F**) Depth-coded Z-stack projection showing the branching of hyphae (blue) and the wall of a peanut testa cell with primary pit fields behind them (yellow). The full set of images, as well as the complete image, are in Appendix SB; (**G**–**I**) Hyphal trajectory featuring branching and transpressoria crossing peanut cells through primary pit fields; (**J**,**K**) Overview of the hyphae and their branches (corresponding to Videos in Appendix A). Abbreviations: arrowheads: branch primordia, ca: cambium, hy: hyphae, pf: primary pit fields, ph: phloem, scp: seed coat parenchyma, tp: transpresorium, vs: vessels, w: walls of *Arachis* testa cells.

**Figure 9 plants-14-01083-f009:**
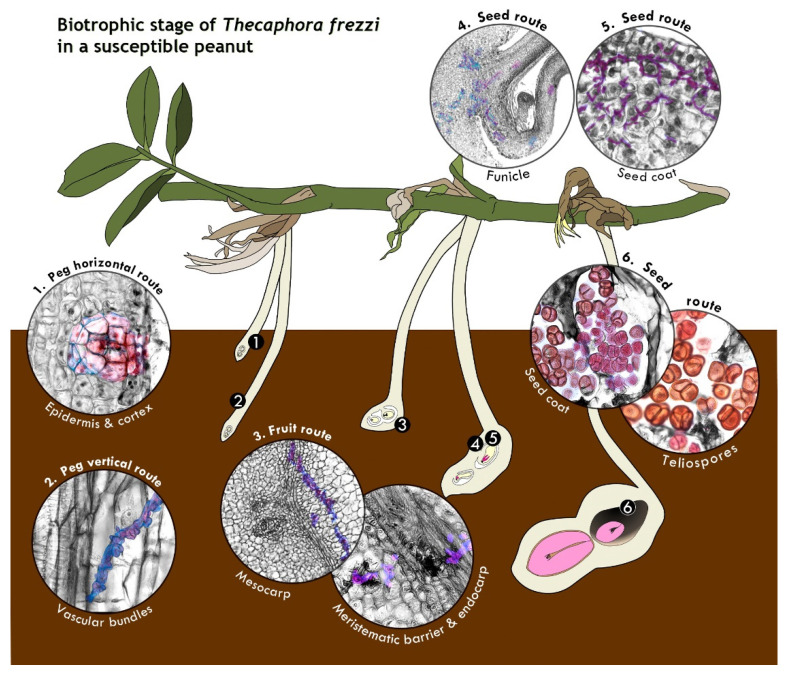
Biotrophic Colonization of *Thecaphora frezzii in Arachis hypogaea* cv. Granoleico. Major events (1–6) of fungal infection are represented, with detailed cellular views in round insets, in different stages (R2–R6) of peanut pod development. The insets show plant tissues in grayscale while fungal structures—micro-nodules, hyphae, and teliospores—are highlighted in color. Images belong to Figure 1, Figure 2, Figure 3, Figure 4, Figure 5, Figure 6, Figure 7 and Figure 8 in Section 2. (1) Fungal entry through peg and micro-nodule at R2-subterranean peg; (2) Hyphae route in vascular tissue at R2-subterranean peg; (3) Hyphae traversing mesocarp, meristematic barrier, and endocarp at R3-medium pod; (4,5) Hyphae in funiculus and seed coat at R3-medium/R3-late pod stages; (6) Teliospores in hypertrophied testa cells and mature teliospores in lysed testa cells at R6-full seed stage.

## Data Availability

The research data are available at http://hdl.handle.net/11336/252434, date deposited 14 January 2025.

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
