# Peer review of "Histopathology of Thecaphora frezzii Colonization: A Detailed Analysis of Its Journey Through Peanut (Arachis hypogaea L.) Tissues"

_plants, 2025, doi:10.3390/plants14071083_

Round 1
Reviewer 1 Report
Comments and Suggestions for Authors
Exploring the Histopathology of Thecaphora frezzii: The 2 Colonizing Journey Through Peanut Tissues (Plants_ 3504998)
Summary: This study investigates the peanut smut, a disease caused by Thecaphora frezzii that has become a significant threat to global peanut production. It aims to fill knowledge gaps about infection, colonization, and sporulation. Using various microscopy techniques, researchers analyzed the susceptible peanut variety Granoleico. They found that T. frezzii enters the host through the peg during the R2-subterranean phase and colonizes the fruit at the R3-stage. Hyphal entry into the seed occurs via the funiculus, leading to extensive seed coat colonization without penetrating the embryo. This triggers hyperplasia and hypertrophy, resulting in teliospore formation and disrupting nutrient translocation, which halts embryo development. The study suggests that examining these structural responses in different peanut genotypes could provide insights into disease susceptibility and resistance mechanisms, which id very important in peanut industry.
Comments:
The study is thoroughly executed with ample experimental plan. However, the flow of the write up should be improved to further improve the essence of the manuscript. For example, the Introduction part must be consolidated as the literature does not flow well and is redundant (line 56 to 127). The first paragraph of the Results section (line 146-156) could start with two sentences and explain in the following subsections. Most of the write ups for the results sections could be made succinct without altering the story flow. Discussion could be also made shorter with more comparative analyses using old literature data from many other plants. Overall, the manuscript is experimentally well presented but improving the writing can further improve the paper.
Comments on the Quality of English Languageno
Author Response
Response:
We appreciate the reviewer's thorough assessment and constructive feedback on the manuscript’s structure. We have aimed to present all data as concisely as possible while ensuring clarity and completeness. In response to this comment, we have summarized certain paragraphs where possible, particularly in the Introduction and Discussion sections, to enhance readability and improve the flow of the manuscript. Additionally, we have carefully reviewed these sections to eliminate redundancies and improve coherence.
However, regarding the Results section, we consider that the level of detail provided is necessary to fully comprehend both the histology of each organ and the modifications induced by the pathogen, as well as to accurately track the pathway of fungal hyphae through different organs, tissues, and cell types. Reducing this level of detail could compromise the clarity and completeness of the findings.
Furthermore, we had to balance multiple reviewer requests during the revision process. One reviewer requested the inclusion of additional references in the Discussion, requiring us to provide proper context for these citations. Another reviewer asked for greater detail in the methodologies used, leading to an expansion of the Materials and Methods section. Given these considerations, we have made careful adjustments to ensure the manuscript remains concise while fulfilling all necessary revisions.
We hope these adjustments provide a well-structured and comprehensive manuscript while maintaining the depth required for a thorough understanding of the pathosystem.

Reviewer 2 Report
Comments and Suggestions for Authors
Dear authors,
This article concerns “Exploring the Histopathology of Thecaphora frezzii: The Colonizing Journey through Peanut Tissues, by Maria Florencia Romero, Sergio Sebastián Samoluk, Guillermo Seijo and Ana Maria Gonzalez. As numerous and precise data, with welcome videos, for this important pathology-agronomy fungus that I read with pleasure, I recommend it for an international audience in this journal, however several details have to be considered by the authors and a major revision is requested.
Please notice that in order to bring a broad audience to this article and to this journal, for specialists and non-specialists, the two points of my comments (at the beginning) are very important (mandatory…) for a suitable value of the article. Minor points are also enhanced at the end of this review.
Sincerely yours,
The two major points are:
- 1 Indicate the number of samples evaluated. In the same way, indicate in material and methods the total number of plants studied for each kind of observations-glass.slides-stainings…, in order to increase the objectivity of your study (LM, SEM, CLSM, measurements...). Do the same when there are values and/or measurements (like in 2.2.3 "10-15 layers" "1-2 layers", in 2.2.4 too; in 2.4 "3.47-4.47 μm"; see for all other parts).
- 2 References already taken in account by the authors are of interest, however checking briefly in web of science WOS and scilit (from mdpi) with some key-words of this manuscript, other (especially recent) references appear and they should be updated and used (if relevant…) in order to sustain much more and provide a larger view of these researches. Among papers are the followings:
[1-5]
- Paredes, J.A.; Sparks, A.H.; Monguillot, J.H.; Rago, A.M.; Molina, J.P.E. Aerial spread of smut spores during peanut harvest. Trop Plant Pathol 2024, 49, 502-514, doi:10.1007/s40858-024-00645-5.
- Guo, Z.; Zhang, J.; Wang, H.; Li, S.; Shao, X.; Xia, L.; Darwish, I.A.; Guo, Y.; Sun, X. Advancing detection of fungal and mycotoxins contamination in grains and oilseeds: Hyperspectral imaging for enhanced food safety. Food chemistry 2025, 470, 142689-142689, doi:10.1016/j.foodchem.2024.142689
- Figueroa, A.C.; Diaz, M.S.; Turco, M.; Trotta, A.F.; Marino, B.; Soria, N.W.; Beltramo, D.M.; Alasino, R.V. Effects of antioxidants on in vitro growth of Thecaphora frezzii. J Appl Microbiol 2024, 135, doi:10.1093/jambio/lxae306.
- Cazon, L.I.; Paredes, J.A.; Miretti, E.; Gonzalez, N.; Suarez, L.; Conforto, C.; Rago, A.M. Decoding peanut smut: A bibliometric analysis of two decades of research progress. Trop Plant Pathol 2024, 49, 547-557, doi:10.1007/s40858-023-00634-0.
- Biswal, A.K.; Ozias-Akins, P.; Holbrook, C.C. Recent Technological Advancements for Identifying and Exploiting Novel Sources of Pest and Disease Resistance for Peanut Improvement. Agronomy-Basel 2024, 14, doi:10.3390/agronomy14123071.
Minor points are:
1 As I am involved in plant taxonomy I am very sensible to correct plant taxa names which make their homogeneity and precision at the international level. In this respect the latin name of each taxon plus the author(s) (not in italics) have to be inserted (at least) the first time they appear in the text (from the beginning of the introduction; for instance for "T. thlaspeos"?). See all other taxa, it is not homogeneous in all parts of your text; find also the author for peanut cultivar or at least the first reference where it was created. Use international Plant Names Index (IPNI) https://www.ipni.org/ or equivalent, for fungi https://www.mycobank.org/ or equivalent.
2 For all figures, although some photos are quite small and not with a lot of free space, indicate the value of the scale bar directly above the scale, it is much more rapid to check, especially with all these various and interesting enlargements of sections.
3 For figure 2c caption, put "bottom right" instead of "below"? Although the stoma is quite nice, if it is not related with the hyphae penetration what is the use of this photo? For E and F, indicate the hyphae. Indicate in full letters the meaning of CLSM (do the same for figure 5), in order to be understood rapidly.
4 In Figure 3A C F and other figures, indicate the hyphae (are they the arrowheads? You indicate “arrows” in your captions).
5 For figure 6C, I cannot see the hyphae of G, rewrite the captions to clarify their homogeneity.
6 For figure 7 boxed area A with star, I do not understand clearly which part it concerns. Moreover, indicate “branched hyphae and thin hyphae”.
7 For figure 8F, I do not understand clearly why it is a 3D reconstruction as it looks like a 2D image.
8 For figure 9, 4 and 5 should be closer.
9 In 3.6, I do not understand clearly the “two characteristics” of the second paragraph, please enhance them with "firstly secondly" or some similar wordings.
10 In 5. you should write hyperplasia instead of hiperplasia?
11 Finally, as a small point, as a specialist of transmission electron microscopy TEM I am, I regret useful ultrathin sections which would allow to clarify (?) the intimate connection(s) between the fungus and the plant as it is provided in some other colonizing fungi studies, however it is not a negative critic for the very good present work and I hope it will be the next step(s) (?) of this very detailed and rich study.
Author Response
Rev2 Dear authors, This article concerns “Exploring the Histopathology of Thecaphora frezzii: The Colonizing Journey through Peanut Tissues, by Maria Florencia Romero, Sergio Sebastián Samoluk, Guillermo Seijo and Ana Maria Gonzalez. As numerous and precise data, with welcome videos, for this important pathology-agronomy fungus that I read with pleasure, I recommend it for an international audience in this journal, however several details have to be considered by the authors and a major revision is requested. Please notice that in order to bring a broad audience to this article and to this journal, for specialists and non-specialists, the two points of my comments (at the beginning) are very important (mandatory…) for a suitable value of the article. Minor points are also enhanced at the end of this review. Sincerely yours, The two major points are: 1 Indicate the number of samples evaluated. In the same way, indicate in material and methods the total number of plants studied for each kind of observations-glass.slides-stainings…, in order to increase the objectivity of your study (LM, SEM, CLSM, measurements...). Do the same when there are values and/or measurements (like in 2.2.3 "10-15 layers" "1-2 layers", in 2.2.4 too; in 2.4 "3.47-4.47 μm"; see for all other parts). 2 References already taken in account by the authors are of interest, however checking briefly in web of science WOS and scilit (from mdpi) with some key-words of this manuscript, other (especially recent) references appear and they should be updated and used (if relevant…) in order to sustain much more and provide a larger view of these researches. Among papers are the followings: [1-5] 1. Paredes, J.A.; Sparks, A.H.; Monguillot, J.H.; Rago, A.M.; Molina, J.P.E. Aerial spread of smut spores during peanut harvest. Trop Plant Pathol 2024, 49, 502-514, doi:10.1007/s40858-024-00645-5. 2. Guo, Z.; Zhang, J.; Wang, H.; Li, S.; Shao, X.; Xia, L.; Darwish, I.A.; Guo, Y.; Sun, X. Advancing detection of fungal and mycotoxins contamination in grains and oilseeds: Hyperspectral imaging for enhanced food safety. Food chemistry 2025, 470, 142689-142689, doi:10.1016/j.foodchem.2024.142689 3. Figueroa, A.C.; Diaz, M.S.; Turco, M.; Trotta, A.F.; Marino, B.; Soria, N.W.; Beltramo, D.M.; Alasino, R.V. Effects of antioxidants on in vitro growth of Thecaphora frezzii. J Appl Microbiol 2024, 135, doi:10.1093/jambio/lxae306. 4. Cazon, L.I.; Paredes, J.A.; Miretti, E.; Gonzalez, N.; Suarez, L.; Conforto, C.; Rago, A.M. Decoding peanut smut: A bibliometric analysis of two decades of research progress. Trop Plant Pathol 2024, 49, 547-557, doi:10.1007/s40858-023-00634-0. 5. Biswal, A.K.; Ozias-Akins, P.; Holbrook, C.C. Recent Technological Advancements for Identifying and Exploiting Novel Sources of Pest and Disease Resistance for Peanut Improvement. Agronomy-Basel 2024, 14, doi:10.3390/agronomy14123071. Minor points are detailed and responded to below. YesCan be improvedMust be improvedNot applicable Does the introduction provide sufficient background and include all relevant references? (x) ( ) ( ) ( ) Is the research design appropriate? (x) ( ) ( ) ( ) Are the methods adequately described? (x) ( ) ( ) ( ) Are the results clearly presented? ( ) (x) ( ) ( ) Are the conclusions supported by the results? (x) ( ) ( ) ( ) —------------------ Response to Reviewer 2 Comments 1. Summary Dear Reviewer: Thank you for your insightful comments and for highlighting key aspects that can enhance the value of our manuscript. We have carefully considered your suggestions and have made every effort to implement the requested changes. Regarding major Point 1, we have made all requested revisions in the Materials and Methods section, specifying the number of samples and plants studied for each observation type and ensuring all quantitative values and measurements are clearly stated. Concerning Point 2, we appreciate the references suggested and the effort to enhance our manuscript. After reviewing the provided citations, we have decided not to incorporate them, as they are not directly relevant to the specific focus of our study. However, we conducted a new literature search to ensure our references remain comprehensive and up to date. As a result, we have incorporated additional citations, particularly in the Discussion section, to further support and expand the contextual analysis of our findings. We hope you understand our decision, as despite this comment being classified under the "mandatory" section, the reviewer explicitly stated that the inclusion of these references should be at our discretion, based on their relevance. Additionally, another reviewer requested the removal of certain citations and the reduction of paragraph length to improve conciseness. Consequently, we had to balance both recommendations while maintaining the clarity and relevance of the references used. Additionally, all the remaining minor comments have been carefully reviewed and corrected where necessary. Detailed responses can be found below, with the corresponding revisions marked in red or tracked in the resubmitted files." We appreciate your understanding and your valuable feedback, which has helped refine our manuscript. Comments 3: As I am involved in plant taxonomy I am very sensible to correct plant taxa names which make their homogeneity and precision at the international level. In this respect the latin name of each taxon plus the author(s) (not in italics) have to be inserted (at least) the first time they appear in the text (from the beginning of the introduction; for instance for "T. thlaspeos"?). See all other taxa, it is not homogeneous in all parts of your text; find also the author for peanut cultivar or at least the first reference where it was created. Use international Plant Names Index (IPNI) https://www.ipni.org/ or equivalent, for fungi https://www.mycobank.org/ or equivalent Response 3: [Thank you for your helpful suggestion. We have added the authors to each taxon the first time they are mentioned. The species for which the author names were added are Thecaphora frezzii Carranza & Lindquist; T. thlaspeos (Beck) Vánky; cv. Granoleico peanut (Reg. No. 7907, El Carmen Nursery, Argentina); Solanum tuberosum L.; Ustilago hordei (Pers.) Lagerh. Arachis L; Thecaphora amaranthicola M. Piepenbr.; U. tritici (Pers.) Rostr. Comments 4: For all figures, although some photos are quite small and not with a lot of free space, indicate the value of the scale bar directly above the scale, it is much more rapid to check, especially with all these various and interesting enlargements of sections. Response 4: Thank you for your helpful suggestion. We have now incorporated the scale bar values directly above the scale in all figures, as recommended, to enhance clarity and facilitate rapid reference.] Comments 5: For figure 2c caption, put "bottom right" instead of "below"? Although the stoma is quite nice, if it is not related with the hyphae penetration what is the use of this photo? For E and F, indicate the hyphae. Indicate in full letters the meaning of CLSM (do the same for figure 5), in order to be understood rapidly. Response 5: We agree with your suggestions. The stoma image and its caption have been removed as recommended. To improve clarity, we have added references to indicate the hyphae in images E and F. Regarding the abbreviation CLSM, we have written it out in full the first time it appears to ensure clarity while avoiding redundancy in subsequent mentions. Comments 6: In Figure 3A C F and other figures, indicate the hyphae (are they the arrowheads? You indicate “arrows” in your captions) Response 6: Thank you for your observation. We have now explicitly indicated the hyphae in Figure 3A, C, F, and other relevant figures for improved clarity. Additionally, we have revised the figure captions to ensure consistency in terminology, specifying whether hyphae are marked with arrows or arrowheads. These changes have been marked in red throughout the manuscript] Comments 7: For figure 7 boxed area A with star, I do not understand clearly which part it concerns. Moreover, indicate “branched hyphae and thin hyphae”. Response 7: [Agree. We have removed the boxed area A with the star from Figure 7 for clarity. Additionally, we have indicated "branched hyphae" and "thin hyphae" as requested.] Comments 8: For figure 8F, I do not understand clearly why it is a 3D reconstruction as it looks like a 2D image. Response 8: [Thank you for bringing this to our attention. Agreed, we have corrected the caption and replaced "(F) 3D reconstruction of photos in E showing the branching of hyphae (blue) and the wall of a peanut testa cell behind them (yellow), note the primary pit fields (full image in Appendix B);" with "(F): Depth-coded Z-stack projection showing the branching of hyphae (blue) and the wall of a peanut testa cell with primary pit fields behind them (yellow). The full set of images, as well as the complete image, are in Appendix B"] Comments 9: For figure 9, 4 and 5 should be closer. Response 9: [Agree. We have adjusted Figure 9 to position images 4 and 5 closer together, improving visual coherence and ease of comparison.] Comments 10: In 3.6, I do not understand clearly the “two characteristics” of the second paragraph, please enhance them with "firstly secondly" or some similar wordings. Response 10: [Agree. We have made the requested clarification in the text, explicitly stating that the two novel structures observed in Thecaphora are branch primordia and transpressorium.] Comments 11: you should write hyperplasia instead of hiperplasia? Response 11: [Agree. We have corrected "hiperplasia" to "hyperplasia" as requested.] Comments 12: 11 Finally, as a small point, as a specialist of transmission electron microscopy TEM I am, I regret useful ultrathin sections which would allow to clarify (?) the intimate connection(s) between the fungus and the plant as it is provided in some other colonizing fungi studies, however it is not a negative critic for the very good present work and I hope it will be the next step(s) (?) of this very detailed and rich study. Response 12: [We sincerely appreciate your valuable insight and your kind words about our study. We fully recognize the significant contributions that ultrathin sections and TEM analysis can provide in elucidating the intimate connections between the fungus and the plant, as demonstrated in other studies on smut fungi. Unfortunately, at this time, we do not have access to the necessary equipment to perform such analyses. Nevertheless, we acknowledge the importance of this approach and certainly consider it an exciting avenue for future research]Final comment to reviewer and editor
Due to the inclusion of new citations as requested, the numbering of references throughout the manuscript has been shifted accordingly. All citations have been adjusted to maintain proper sequence and consistency within the text.
We would like to inform you that all figures, including those within the main text as well as Supplementary Figures 1 and 2, have been modified according to the reviewers' comments requesting the addition of scale bars. As a result, we are re-submitting the updated figures with these adjustments incorporated.

Reviewer 3 Report
Comments and Suggestions for Authors
Journal Plants (ISSN 2223-7747)
Manuscript ID plants-3504998
Type Article
Title Exploring the Histopathology of Thecaphora frezzii: The Colonizing Journey Through Peanut Tissues
Authors Maria Florencia Romero * , Sergio Sebastián Samoluk , Guillermo José Seijo , Ana Maria Gonzalez *
Section Plant Development and Morphogenesis
Special Issue Anatomical, Ontogenetic, and Embryological Studies of Plants
Dear Editor
The text of manuscript ID plants-3504998 “Exploring the Histopathology of Thecaphora frezzii: The Colonizing Journey Through Peanut Tissues”
requires substantive and editorial revision.
I provide comments on the text below.
With best regards
Reviewer
Title
1.
Please think about the topic of your manuscript.
2.
Eliminate in the subject line „research „
3.
Please include the Latin name of the species in the subject line of the manuscript
Abstract
4.
Please complete which „....may provide key insights into...”
Introduction
- 40-65.
5.
Please complete the values instead of giving classes of compounds (it has been known for years). When citing five scientific publications, the reader expects factual knowledge.
6.
Please state the health-promoting benefits instead of in which form they are on the market (specify which biological model, which biologically active compound , dose , metabolic process beneficial to the body, etc.).
7.
Please describe in detail the problem of infection at the cellular level „:... caused by the fungus Thecaphora frezzii Lindquist and Carranza...”.
8.
Please complete the estimated losses specifically in numbers in relation to the relevant region „...significantly reducing the peanut yield...”.
Please complete details of specific information and estimated yield reduction instead of general statements (known for years) „The severity of this disease varies depending on environmental conditions and agricultural practices, and can produce yield losses higher than 30% [9,14,15].”
9.
When citing two or three publications, please complete the specific information derived from the publications in question with close reference to the subject of the manuscript. „
- 66-83.
10.
Two sentences must not form a subject paragraph
11.
Please complete what (dosage, method of application, etc. ) „...including the use of fungicides...”.
12.
The examples given „....wheat, maize, and barley....” are distant in systematic position, please complete the reference to the manuscript topic.
- 84-113
13.
Please complete what „...and the signals required...”.
14.
The examples/descriptions given should be closely related to the topic of the manuscript.
- 114-144.
15.
- 114-120 School viewer, not in this place and not with this level of information. Suggests inserting thematic subsections in the introduction, arranged chronologically so that one follows from the other.
16.
Please reinforce the rationale for the research undertaken by indicating what has been done and what gap exists in terms of the research presented.
Results
17.
- 146- 157 Not here, please modify the text and move to the appropriate subsection.
- 159-202.
18.
The results are not the place to cite the literature, nor is the introduction. Please modify this.
19.
Please reword the text eliminating the five-fold citation „....Figure 1A-B....”
20.
Figure 1 Photographs should be arranged from lowest to highest magnification.
21.
Figure 1. The bar should be applied to each photograph including the figure and name, instead of in the description under the photograph.
- 203 - 206
22.
A single sentence must not constitute a subsection.
- 208 - 298.
23.
See note above „(Figure 2D-G).”
24.
Figure 2, 3. Please enter the cell designation on the sections.
- 299-352.
25.
Figure 4. Please see comment above, cell markings in photographs should be completed.
26.
Please eliminate multiple repetition of figure designations e.g. „Figure 4A-C).”
- 319-533
27.
Please apply correction to text and figures as per comments above on results.
Disscusion
28.
Please introduce more references to the research presented in each subsection.
29.
Figure 9. is there????
Materials and Methods
30.
Please provide details in the description of light, confocal and scanning microscopy in the following steps e.g.: fixation of material, dehydration, preparation of semi-thin slides, drying, sputtering etc. Please add: concentrations, stains and other reagents. This is a very specific method of preparing microscope slides
31.
Please state how the observation scheme was adopted to eliminate accidental shots.
32.
Please check the „12 μm thickness”.
References
33.
Please check each item of literature as indicated for the authors, e.g. item 5, 15, 44.
PROOFREADING THE MANUSCRIPT TEXT
In the text correction, please indicate in red the changes made.
35.
In the response letter to the Reviewer”s comments, please provide a response (in red) to each comment and paste the passage inserted in the manuscript into the comment in question.

Author Response
Response to Reviewer 3 Comments
1. Summary
We appreciate the reviewer’s thorough evaluation and constructive feedback on our manuscript. We have carefully addressed the majority of the comments by implementing the suggested modifications throughout the text. These revisions include refining the title for clarity, enhancing the Introduction and Discussion sections for conciseness and coherence, incorporating additional references for a broader comparative analysis, and providing greater detail in the Materials and Methods section to improve methodological transparency. Furthermore, we have streamlined certain paragraphs, eliminated redundancies, and improved the logical flow of the manuscript while maintaining the necessary level of detail to support the study’s objectives.
However, some requests were not implemented due to the need to balance multiple reviewer suggestions. For example, while one reviewer requested a more concise Discussion, another recommended the inclusion of additional references that required proper contextualization. Similarly, in the Materials and Methods section, a third reviewer asked for a higher level of detail regarding microscopy techniques, which necessitated expansion in that section. Additionally, in some cases, the requested information was already present in the text, either in a different section or in an appropriate format aligned with the manuscript's focus. In such instances, we have provided explanations for our decisions.
We hope the reviewer understands the rationale behind these adjustments, and we truly appreciate the insightful comments that have contributed to improving the clarity and depth of our manuscript.
Title
Comments 1.Please think about the topic of your manuscript. Please include the Latin name of the species in the subject line of the manuscript
Response 1.3.: [We have reconsidered and modified the title as requested, changing "Exploring the Histopathology of Thecaphora frezzii: The Colonizing Journey Through Peanut Tissues" to "Histopathology of Thecaphora frezzii Colonization: A Detailed Analysis of Its Journey Through Peanut (Arachis hypogaea) Tissues." This revision incorporates the scientific name of peanut and provides a more precise and descriptive title. ]
Comments 2.Eliminate in the subject line „research „
Response 2: [After carefully reviewing both the title and the abstract, we confirm that the word "research" is not present in subject section.]
Abstract
Comments 4. Please complete which „....may provide key insights into...”
Response 4: [Agree. We have addressed the request by replacing the phrase "may provide key insights into..." in the abstract” by “may provide key insights into the anatomical barriers and defense mechanisms that determine disease susceptibility and resistance, ultimately contributing to the development of resistant cultivars”.]
Introduction 40-65.
Comments 5. Please complete the values instead of giving classes of compounds (it has been known for years). When citing five scientific publications, the reader expects factual knowledge.
Response 5: [We have added the general range percentages of protein (15 to 28%), oil (45-56%) as requested. However, including detailed data on vitamin and micronutrient content would significantly extend the text with information already available in another article. Additionally, another reviewer requested shortening the manuscript, so we aimed to balance completeness with conciseness. Thank you for your understanding.]
Comments 6. Please state the health-promoting benefits instead of in which form they are on the market (specify which biological model, which biologically active compound, dose, metabolic process beneficial to the body, etc.).
Response 6: [Agree. We incorporated your suggestion and have revised the text to focus on the nutritional density and health-promoting components of peanuts rather than their market forms. The corrected sentence is as follow: “.. and a variety of healthy micronutrients and bioactive compounds, peanuts provide the highest protein content among commonly consumed snack nuts and serve as a rich source of heart-healthy monounsaturated oil, contributing to their nutritional value and potential health benefits”]
Comments 7.Please describe in detail the problem of infection at the cellular level „:... caused by the fungus Thecaphora frezzii Lindquist and Carranza...”.
Response 7: [The only known detail about the infection is already stated in the following sentences: "Pods of plants infected by T. frezzii may exhibit hypertrophy and abnormal shapes, but the most distinguishing symptom is the production of teliospores within the pods, which occurs in the later stages of infection [10]. These teliospores accumulate, replacing the kernel tissue with a smutted mass." Additionally, there is a rough mention of the tissues which hyphae colonize (mentioned in discussion), but no information is available on the specific cellular mechanisms involved in the infection process. The lack of this information motivated one of the main objectives of the manuscript]
Comments 8. Please complete the estimated losses specifically in numbers in relation to the relevant region „...significantly reducing the peanut yield...”.
Response 8: [We would like to clarify that the specific numerical estimation of yield losses is already provided in the following paragraph of the manuscript. The values of losses are given in the subsequent sentence of manuscript as a percentage of the estimated yield in the affected plots. Information is only available for the main production area in Argentina. Anyway it was edited for clarity. The updated sentence is as follow ”The severity of this disease varies depending on environmental conditions and agricultural practices, and fields with highest incidence and severity show yield losses up to 35%”]
Comments 8 bis. Please complete details of specific information and estimated yield reduction instead of general statements (known for years) „The severity of this disease varies depending on environmental conditions and agricultural practices, and can produce yield losses higher than 30% [9,14,15].
Response 8 bis: The percentage of yield loss (up to 35%) provides an indication that allows the reader to estimate the disease's impact on peanut production. Since the purpose of this manuscript is not to present an epidemiological assay, we believe that incorporating the requested details would divert the reader's focus from the main objective of the study. For more details on the epidemiology of the disease, see reviews 9, 14, 15 in the literature.
Comments 9. When citing two or three publications, please complete the specific information derived from the publications in question with close reference to the subject of the manuscript. „
Response 9: [In this paragraph, each time more than one citation is used, it was done to refer to the same topic. The intention was to bring higher support for the statement given in the introduction. Additionally, another reviewer requested that this section be shortened. To balance these recommendations, we have maintained the citations to ensure proper scientific backing while optimizing the text for conciseness.]
83. 66-83.
Comments 10. Two sentences must not form a subject paragraph
Response 10: [Agree. This suggestion was corrected; in two instances, a period was inadvertently included.]
Comments 11. Various strategies have been explored to manage peanut smut with poor to moderate and variable results, including the use of fungicides [9,13,15]. Please complete what (dosage, method of application, etc. ) „...including the use of fungicides...”.
Response 11: [The manuscript does not address the chemical controls of smut and their effectiveness; it merely provides a reference to highlight that the most effective solution to the disease so far comes from the introgression of alleles from wild Arachis species or local peanut landraces. The details regarding dosage and method of application (detailed in excellent reviews (Rago et al 2017, Paredes et al 2024) do not add essential information to this study, as fungicide use was only mentioned to provide context on the problem. Additionally, another reviewer requested shortening the manuscript, so this information was not included. Thank you for your understanding.]
Comments 12. The examples given „....wheat, maize, and barley....” are distant in systematic position, please complete the reference to the manuscript topic.
Response 12: [We have revised the text to clarify the relevance of wheat, maize, and barley as hosts of other Ustilaginales smut fungi, while also emphasizing how Thecaphora frezzii differs in host specificity and infection strategy. The modified text highlights that, unlike cereal-infecting smuts that primarily colonize floral tissues, T. frezzii invades through the peg and colonizes the developing fruit, following a distinct infection route. This adjustment ensures a clearer connection to the manuscript's focus while maintaining relevant context. ]
1. 84-113
Comments 13. Please complete what „...and the signals required...”.
Response 13: [Agree. To enhance clarity, we have revised the sentence by specifying the type of signals involved. The updated version now includes the terms "endogenous plant signals" to explicitly indicate their origin and relevance.]
Comments 14. 114-144. The examples/descriptions given should be closely related to the topic of the manuscript.
Response 14: All the examples are closely related to the topic, specifically the biotrophic life cycle and histopathology of different smuts, and they are necessary in the discussion to compare different pathways of hyphal colonization. The comparison of T. frezii with most of the mentioned smuts has been presented in various reference reviews on smuts, such as Arias et al. (2021) and Van der Linde et al. (2021). For this reason, we consider their inclusion in the introduction to be appropriate.]
Comments 15.114-120 School viewer, not in this place and not with this level of information. Suggests inserting thematic subsections in the introduction, arranged chronologically so that one follows from the other.
Response 15: [We cannot understand what the suggestion in the first sentence refers to regarding School Viewer. Additionally, we are not sure that the second sentence pertains to our article, as inserting thematic subsections in the introduction arranged chronologically is not mandatory for a scientific paper.]
Comments 16.Please reinforce the rationale for the research undertaken by indicating what has been done and what gap exists in terms of the research presented.
Response 16: [Agree. The sentence was clarified incorporating the remaining gaps of knowledge. The sentence was corrected as follows: “Despite advancements in characterizing —identifying the peg as the primary entry point for hyphae, the vascular bundles as the conduit for invasion, and the fruit as the site of sporulation, ultimately replacing the kernels [8, 35–38], substantial gaps remain in our understanding of the infection pathways, anatomical responses, and mechanisms of colonization and sporulation in peanut tissues.”]
Results
Comments 17.146- 157 Not here, please modify the text and move to the appropriate subsection.
Response 17: [To address this comment, we have modified the text accordingly. The details of the reproductive stages of peanut development have been integrated into the Introduction to provide relevant background information in the context of the study’s objectives. This ensures that the developmental stages are introduced earlier, establishing their relevance to the timing of T. frezzii colonization. Additionally, in section 2.1. Developmental stages of peanut fruit and seed associated with T. frezzii colonization, we have retained the description of the reproductive stages but without the citation, ensuring the section remains focused on its primary aim—describing fungal invasion in relation to host development.]
1. 159-202.
Comments 18. The results are not the place to cite the literature, nor is the introduction. Please modify this.
Response 18: [We have removed the only citation from the Results section to align with the reviewer's recommendation. However, we believe the phrase "nor is the introduction" may have been a typographical error, as citations are essential in the Introduction to provide context, justify the study, and reference key publications, as specified in the journal's author instructions: "The introduction should briefly place the study in a broad context and highlight why it is important… The current state of the research field should be reviewed carefully and key publications cited."]
Comments 19. Please reword the text eliminating the five-fold citation „....Figure 1A-B....”
Response 19: [Agree, we have revised the text to eliminate the repeated citation of "Figure 1A-B" while maintaining clarity and readability. ]
Comments 20. Figure 1 Photographs should be arranged from lowest to highest magnification.
Response 20: [The photographs in Figure 1 have been arranged in the order in which they are cited in the text, ensuring a logical sequence that aligns with the manuscript's description. Additionally, the images were organized based on developmental stages, with the youngest stages in photo A, where they are more clearly visible, and the later stages in photo B. This arrangement was prioritized over organizing them strictly by magnification level to maintain coherence with the narrative. Due to space constraints on the figure layout, photo C was placed next to photo A. For these reasons, we have decided to keep the current arrangement.]
Comments 21. Figure 1. The bar should be applied to each photograph including the figure and name, instead of in the description under the photograph.
Response 21: [Thank you for your suggestion. We have implemented this change in all figures, ensuring that the scale bar is directly applied to each photograph along with the corresponding figure label, rather than placing it in the description.]
1. 203 - 206
Comments 22.A single sentence must not constitute a subsection.
Response 22: [We would like to clarify that the initial sentence was not intended to function as an independent subsection but rather as an introductory statement to guide the reader through the upcoming divisions of the section. Given that the section is subdivided into more specific parts, this introductory sentence was meant to provide context and a logical transition, helping the reader understand the structure and flow of the information. We kindly ask the reviewer to consider the numbering of the subsections, which indicates that this sentence is part of a larger structured section rather than a standalone subsection.]
298. 208 - 298.
Comments 23. See note above „(Figure 2D-G).”
Response 23: [Agreed, we have revised the text to eliminate the repeated citation.]
Comments 24.Figure 2, 3. Please enter the cell designation on the sections.
Comments 25.Figure 4. Please see comment above, cell markings in photographs should be completed
Comments 27.Please apply correction to text and figures as per comments above on results.
Response 24-25-27: [The cell designation is described in detail in the following paragraphs. To address the reviewer's comment, we have removed that reference from the subtitle to avoid redundancy ]
352. 299-352.
Comments 26. Please eliminate multiple repetition of figure designations e.g. “Figure 4A-C).”
Response 26: [Agree, we have revised the text to eliminate the repeated citation of "Figure 4A-C" while maintaining clarity and readability. ]
Disscusion
Comments 28. Please introduce more references to the research presented in each subsection.
Response 28: [We have added additional references throughout the subsections to strengthen the discussion and provide further context to the research presented. Additionally, we have incorporated some older references that, while already cited in the works previously mentioned, have now been explicitly included in response to the reviewer's recommendation. This ensures a more comprehensive citation of relevant literature and enhances the depth of the discussion. ]
Comments 29. Figure 9. is there????
Response 29: [Yes, Figure 9 is intentionally included in the Discussion section as a way to summarize key findings and facilitate their interpretation within the broader context of the study. The inclusion of figures in the Discussion serves to visually reinforce the results being analyzed, making it easier for the reader to follow the arguments and comparisons presented. This approach aligns with standard scientific practice, where figures help to integrate and contextualize findings within the discussion, enhancing clarity and comprehension]
Materials and Methods
Comments 30. Please provide details in the description of light, confocal and scanning microscopy in the following steps e.g.: fixation of material, dehydration, preparation of semi-thin slides, drying, sputtering etc. Please add: concentrations, stains and other reagents. This is a very specific method of preparing microscope slides
Response 30: [We have incorporated detailed information on the preparation steps for light microscopy (LM), confocal laser scanning microscopy (CLSM), and scanning electron microscopy (SEM) in their respective sections, including dehydration, drying, sputtering, as well as the concentrations of stains and reagents used. To avoid redundancy, we have kept only the type of fixation in the initial paragraph, as it was the same for all subsequent studies (LM, CLSM, and SEM).
In response, the attached documents contain the revised texts incorporating the requested changes.
Comments 31. Please state how the observation scheme was adopted to eliminate accidental shots.
Response 31: We confirm that all observations and photographic captures were performed manually, without the use of an automated system, ensuring that no accidental shots were taken. To clarify this aspect, we have added the following sentence to the Materials and Methods section: "All observations and photographic captures were performed manually, without the use of an automated system, ensuring that no accidental shots were taken."]
Comments 32. Please check the „12 μm thickness”.
Response 32: [We have reviewed the thickness specification and confirm that "12 μm" is the correct value used for the sections of material embedded in paraffin. No changes were made to the text.]
References
Comments 33. Please check each item of literature as indicated for the authors, e.g. item 5, 15, 44.
Response 33: [We have carefully checked each item in the reference list, including items 5, 15, and 44, as indicated by the reviewer. Minor errors were identified and corrected to ensure accuracy and compliance with the journal’s guidelines.]
PROOFREADING THE MANUSCRIPT TEXT
34. In the text correction, please indicate in red the changes made.
35.In the response letter to the Reviewer”s comments, please provide a response (in red) to each comment and paste the passage inserted in the manuscript into the comment in question.
Response: Both requests have been taken into account. In the revised manuscript, all changes have been tracked, with deleted words or paragraphs marked accordingly and newly added text highlighted in red. Additionally, in the response letter, we have provided a response to each comment, including the revised passage from the manuscript, also marked in red. This ensures full transparency in the modifications made.
Final comment to reviewer and editor
Due to the inclusion of new citations as requested, the numbering of references throughout the manuscript has been shifted accordingly. All citations have been adjusted to maintain proper sequence and consistency within the text.
We would like to inform you that all figures, including those within the main text as well as Supplementary Figures 1 and 2, have been modified according to the reviewers' comments requesting the addition of scale bars. As a result, we are re-submitting the updated figures with these adjustments incorporated.

Round 2
Reviewer 2 Report
Comments and Suggestions for Authors
Dear authors,
I read this second version with pleasure, including your remarks, for this work very well done. I have just one very minor suggestion:
- In your correction of point 6, although I am not English native speaker, since there is actually no "place", “in the second place” does not seem very accurate, "secondly" seems better.
Yours very sincerely,
Author Response
Dear Reviewer,
Thank you very much for your kind words and for taking the time to read our revised version. We truly appreciate your thoughtful feedback and your valuable suggestion regarding point 6.
We agree that "secondly" is a more appropriate choice and will make this adjustment accordingly.
Your careful review and constructive comments have greatly contributed to improving our manuscript.
Best regards,
The authors

Reviewer 3 Report
Comments and Suggestions for Authors
Journal Plants (ISSN 2223-7747)
Manuscript ID plants-3504998
Type Article
Title Exploring the Histopathology of Thecaphora frezzii: The Colonizing Journey Through Peanut Tissues
Authors Maria Florencia Romero * , Sergio Sebastián Samoluk , Guillermo José Seijo , Ana Maria Gonzalez *
Section Plant Development and Morphogenesis
Special Issue Anatomical, Ontogenetic, and Embryological Studies of Plants
Dear Editor
In its current form, the manuscript cannot proceed to further editorial stages.
The authors are arguing with the Reviewer's comments instead of improving the text.
I emphasise, publications in an international journal with high parameters (‘Impact Factor: 4.0 202); 5-Year Impact Factor: 4.4 (2023)’) should meet the accepted REVIEW standards.
I give below some comments in addition to the many unrealised suggestions in the review - round one.
With best regards
Reviewer
1.
In the title of the manuscript, please complete the abbreviation of the discoverer of the species „... (Arachis hypogaea)...”.
2.
Please make a correction e.g. l. 63,169, 172, 328, 487. 797, 907, 925,
3.
Photograph 1.
According to accepted standards, the arrangement of photographs in a figure should follow increasing magnification.
First photograph C
Second photograph A
Third photograph B
Please change this.
4.
- 194 - 198 One sentence cannot be a subsection.
5.
Please eliminate double quotation in adjacent tasks of the same figure „...(Figure 3A)...(Figure 3A)...” , „...(Figure 5A-C). ...” etc. Please replace this elsewhere in the manuscript also.
6.
Please complete the labels of the cells that make up the corresponding tissue on each phonograph in the figure. Not all readers have knowledge of plant anatomy.
Figure 8 description please format the text.
Discussion is not the place to quote figures.
9.
Please check the compliance of the citation of literature items with the authors” guidelines, e.g. item: 5, 21, 39, 44 etc. There are still errors in REFERENCES.
Comments on the Quality of English LanguageJournal Plants (ISSN 2223-7747)
Manuscript ID plants-3504998
Type Article
Title Exploring the Histopathology of Thecaphora frezzii: The Colonizing Journey Through Peanut Tissues
Authors Maria Florencia Romero * , Sergio Sebastián Samoluk , Guillermo José Seijo , Ana Maria Gonzalez *
Section Plant Development and Morphogenesis
Special Issue Anatomical, Ontogenetic, and Embryological Studies of Plants
Dear Editor
In its current form, the manuscript cannot proceed to further editorial stages.
The authors are arguing with the Reviewer's comments instead of improving the text.
I emphasise, publications in an international journal with high parameters (‘Impact Factor: 4.0 202); 5-Year Impact Factor: 4.4 (2023)’) should meet the accepted REVIEW standards.
I give below some comments in addition to the many unrealised suggestions in the review - round one.
With best regards
Reviewer
1.
In the title of the manuscript, please complete the abbreviation of the discoverer of the species „... (Arachis hypogaea)...”.
2.
Please make a correction e.g. l. 63,169, 172, 328, 487. 797, 907, 925,
3.
Photograph 1.
According to accepted standards, the arrangement of photographs in a figure should follow increasing magnification.
First photograph C
Second photograph A
Third photograph B
Please change this.
4.
- 194 - 198 One sentence cannot be a subsection.
5.
Please eliminate double quotation in adjacent tasks of the same figure „...(Figure 3A)...(Figure 3A)...” , „...(Figure 5A-C). ...” etc. Please replace this elsewhere in the manuscript also.
6.
Please complete the labels of the cells that make up the corresponding tissue on each phonograph in the figure. Not all readers have knowledge of plant anatomy.
Figure 8 description please format the text.
Discussion is not the place to quote figures.
9.
Please check the compliance of the citation of literature items with the authors” guidelines, e.g. item: 5, 21, 39, 44 etc. There are still errors in REFERENCES.
Author Response
Dear Reviewer,
We greatly appreciate your thorough and constructive evaluation of our manuscript. Your detailed comments have significantly contributed to improving the manuscript, and we have carefully considered and addressed each of your suggestions. It is not, and has never been, our intention to dispute your expert opinions. Instead, we have sought to follow the journal's guidelines, which explicitly permit authors to respectfully decline suggested changes if adequately justified.
Below, we detail our specific responses to each of your valuable comments.
Comment 1. In the title of the manuscript, please complete the abbreviation of the discoverer of the species „... (Arachis hypogaea)...”.
Response 1: [We add to the abbreviation of the author of Arachis hypogaea in the title]
Comment 2. Please make a correction e.g. l. 63,169, 172, 328, 487. 797, 907, 925,
Comment 2. L. 63 and following. Comment (Round 1): Please describe in detail the problem of infection at the cellular level „:... caused by the fungus Thecaphora frezzii Lindquist and Carranza...” // Please complete the estimated losses specifically in numbers in relation to the relevant region „...significantly reducing the peanut yield...”// Please complete details of specific information and estimated yield reduction instead of general statements (known for years) „The severity of this disease varies depending on environmental conditions and agricultural practices, and can produce yield losses higher than 30% [9,14,15].
Response (Round 2): We have incorporated specific details present in the cited references, and additional new references were included to address the reviewer's requests. Following your suggestions, we have expanded the paragraph by including detailed information from the literature regarding cellular alterations caused by the fungus. Specifically, we incorporated current knowledge about the infection process, clearly mentioning the affected tissues and related histological changes. This information was previously located further ahead in the introduction and has now been consolidated into this section for clarity. Additionally, we have included specific quantitative data on yield reductions, explicitly indicating significant losses under defined environmental and agricultural conditions relevant to the main peanut-producing regions.
Comment 2: L. 169, 172, 328, 487; and Comment 5: Please eliminate double quotation in adjacent tasks of the same figure „...(Figure 3A)...(Figure 3A)...” , „...(Figure 5A-C). ...” etc. Please replace this elsewhere in the manuscript also.
Comments (round 1) it was requested: Please reword the text eliminating the five-fold citation „....Figure 1A-B....” // 319-533. Please apply correction to text and figures as per comments above on results. // See note above „(Figure 2D-G).”
Response: [We have carefully reviewed the entire manuscript and removed several figure references wherever possible. However, in certain cases, we retained some references that might appear repetitive because they are necessary for clarity and readability. Some figures illustrate multiple structures or developmental stages, which are described in different paragraphs. Although these features appear within the same images, guiding the reader to the relevant visual elements at appropriate points in the text is essential. This ensures that key observations are easily located and understood. Therefore, while we minimized redundancy, we maintained figure citations where they provide crucial context for different aspects of the study.]
Comment 2. L. 797,
Response: In the original manuscript, the mentioned line was the last sentence of the Discussion section. However, in the revised version, it is now part of the discussion under Section 3.6: Hyphal and Teliospore Morphology. We believe the reviewer may be referring to this comment from the first round of review: Comments 28. Please introduce more references to the research presented in each subsection.
In our initial response, we have already addressed this comment. We have added additional references throughout the subsections to strengthen the discussion and provide further context to the research presented. Additionally, we have incorporated some older references that, while already cited in the works previously mentioned, have now been explicitly included in response to the reviewer's recommendation. This ensures a more comprehensive citation of relevant literature and enhances the depth of the discussion.
Comment 2. L. 907, 925,
Response: In line 907, it previously stated: "..... stained with 0.01% Calcofluor White M2R in distilled water, according to the manufacturer's instructions, and then mounted in pure glycerol”
We have added more details about the manufacturer's methodology regarding the use of the stain.
Comment 3. Photograph 1. According to accepted standards, the arrangement of photographs in a figure should follow increasing magnification. Please change this.
Response 3: [We have rearranged the photographs in Figure 1]
Comment 4.194 - 198 One sentence cannot be a subsection.
Response 4: [We have adjusted the placement of the paragraph to better align with this request].
Comment 6. Please complete the labels of the cells that make up the corresponding tissue on each phonograph in the figure. Not all readers have knowledge of plant anatomy.
Response 6: [We add references indicating the cell types in figures, that guide the reader to recognise the cells. Additionally, we have reviewed and corrected the figure legends where necessary.]
Comment 7. Figure 8 description please format the text.
Response 7: [We have formatted the text in the description of Figure 8 as requested. Additionally, we reviewed the previous figure descriptions to ensure consistency and correct any punctuation details.]
Comment 8: Discussion is not the place to quote figures.
Response 8: [Thank you again for highlighting this point. While we fully understand your concern regarding the placement of figures in the Discussion section, we respectfully reiterate our initial response (in Round 1). We intentionally included Figure 9 in the Discussion to effectively summarize key findings and facilitate their interpretation within the broader context of the study, an approach consistent with numerous examples published previously in the journal. However, we fully acknowledge your perspective and, rather than debating your position further, we defer to the editor's judgment on this matter, as we understand the editorial guidelines do provide authors with the option to include figures in the Discussion when justified.]
Comment 9. Please check the compliance of the citation of literature items with the authors” guidelines, e.g. item: 5, 21, 39, 44 etc. There are still errors in REFERENCES.
Response 9: We have revised and corrected the indicated literature citations (e.g., items: 5, 21, 39, 44) to comply with the journal’s author guidelines. Due to the inclusion of new citations as requested, the numbering of references throughout the manuscript has been shifted accordingly. All citations have been adjusted to maintain proper sequence and consistency within the text.

Round 3
